# Longitudinal discontinuities in riverine greenhouse gas dynamics generated by dams and urban wastewater

Hyojin Jin[1], Tae Kyung Yoon[2], Most Shirina Begum[1], Eun-Ju Lee[3], Neung-Hwan Oh[3], Namgoo Kang[4, 5], Ji-Hyung Park[1]

[1]Department of Environmental Science and Engineering, Ewha Womans University, Seoul 03760, Republic of Korea
[2]Division of Environmental Strategy, Korea Environment Institute, Sejong 30147, Republic of Korea
[3]Graduate School of Environmental Studies, Seoul National University, Seoul 08826, Republic of Korea
[4]Center for Gas Analysis, Korea Research Institute of Standards and Science, Daejeon 34113, Republic of Korea
[5]Science of Measurement, University of Science and Technology, Daejeon 34113, Republic of Korea

*Correspondence to*: Ji-Hyung Park (jhp@ewha.ac.kr); Namgoo Kang (nkang@kriss.re.kr)

**Abstract.**

Surface water concentrations of $CO_2$, $CH_4$, and $N_2O$ have rarely been measured simultaneously in river systems modified by human activities, contributing to large uncertainties in estimating global riverine emissions of greenhouse gases (GHGs). Basin-wide surveys of the three GHGs were combined with a small number of measurements of C isotope ratios in dissolved organic matter (DOM), $CO_2$, and $CH_4$ in the Han River basin, South Korea to examine how longitudinal patterns of the three gases and DOM are affected by four cascade dams along a middle section of the North Han River (hereafter termed "middle reach") and treated wastewater discharged to the lower Han River ("lower reach") traversing the Seoul metropolitan area. Monthly monitoring and two-season comparison were conducted at 6 and 15 sites, respectively, to measure surface water gas concentrations and ancillary water quality parameters including concentrations of dissolved organic carbon (DOC) and optical properties of DOM. The basin-wide surveys were complemented with a sampling cruise along the lower reach and synoptic samplings along an urban tributary delivering effluents from a large wastewater treatment plant (WWTP) to the lower reach. The levels of $pCO_2$ were relatively low in the middle reach (51–2465 µatm), particularly at the four dam sites (51–761 µatm), compared with those found in the largely forested upper basin with scattered patches of croplands (163–2539 µatm), the lower reach (78–11298 µatm,) and three urban tributaries (2120–11970 µatm). The upper and middle reaches displayed generally low concentration ranges of $CH_4$ and $N_2O$, with some local peaks influenced by agricultural runoff and impoundments. By comparison, the lower reach often exhibited exceptionally high concentrations of $CH_4$ (1.2–15766 nmol $L^{-1}$) and $N_2O$ (7.5–1396 nmol $L^{-1}$), which were significantly correlated with different sets of variables such as DO and $PO_4^{3-}$ for $CH_4$ and $NH_4^+$ and $NO_3^-$ for $N_2O$. Down-river increases in the levels of DOC and optical properties such as fluorescence index (FI) and protein-like fluorescence indicated an increasing DOM fraction of anthropogenic and microbial origin. The concentrations of the three GHGs and DOC were similar in magnitude and temporal variation at a WWTP discharge and the receiving tributary, indicating a disproportionate contribution of the WWTP effluents to the tributary gas and DOC exports to the lower reach. The values of $\delta^{13}C$ in surface water $CO_2$ and $CH_4$ measured during the sampling cruise along the lower reach, combined with $\delta^{13}C$

and $\Delta^{14}C$ in DOM sampled across the basin, implied a strong influence of the wastewater-derived gases and aged DOM delivered by the urban tributaries. The downstream enrichment of $^{13}C$ in $CO_2$ and $CH_4$ suggested that the spatial distribution of these gases across the eutrophic lower reach may also be constrained by multiple concomitant processes including outgassing, photosynthesis, and $CH_4$ oxidation. The overall results suggest that dams and urban wastewater may create longitudinal discontinuities in riverine metabolic processes leading to large spatial variations in the three GHGs correlating with different combinations of DOM properties and nutrients. Further research is required to evaluate the relative contributions of anthropogenic and in-stream sources of the three gases and DOM in eutrophic urbanized river systems and constrain key factors for the contrasting impoundment effects such as autotrophy-driven decreases in $pCO_2$ and in-lake production of $CH_4$ and $N_2O$.

## 1 Introduction

A growing number of studies have provided a wide range of estimates for the global greenhouse gas (GHG) emissions from inland waters (Cole et al., 2007; Bastviken et al., 2011; Raymond et al., 2013; Lauerwald et al., 2015; Stanley et al., 2016; Marx et al., 2017). Conceptual frameworks have also been proposed to incorporate anthropogenic perturbations as a critical driver of riverine biogeochemical processes in human-impacted river systems (Kaushal et al., 2012; Regnier et al., 2013; Park et al., 2018). Recent studies in large river systems such as the Amazon and Congo have identified wetlands as previously unrecognized sources of $CO_2$ and organic matter for rivers (Abril et al., 2014; Borges et al., 2015). However, these recent efforts have been hampered by data scarcity and inequality and inadequate consideration of multiple GHGs co-regulated by a wide range of concurrent environmental changes including anthropogenic perturbations. While the surface water partial pressure of $CO_2$ ($pCO_2$) has been calculated from available water quality data such as pH and alkalinity to estimate $CO_2$ emissions from a wide range of inland water systems (Lauerwald et al., 2013; Raymond et al., 2013), substantial overestimation of $pCO_2$ can occur in acidic, organic-rich inland waters due to the contribution of organic acids to alkalinity and the limited carbonate buffering (Abril et al., 2015). Another critical issue is the lack of reliable measurements of $pCO_2$ in many large river systems across Asia and Africa (Raymond et al., 2013; Lauerwald et al., 2015). Furthermore, three major GHGs ($CO_2$, $CH_4$, and $N_2O$) have rarely been measured simultaneously across different components of river systems except for a small number of large, 'pristine' rivers such as the Amazon (Richey et al., 1988) and the Congo (Borges et al., 2015; Teodoru et al., 2015) or highly human-impacted systems (Smith et al., 2017; Wang et al., 2017b; Borges et al., 2018). While global river systems are now subject to multiple environmental stresses, including water pollution, impoundments, and climate change, most research efforts have addressed these multiple stresses separately. A more integrative approach addressing multiple GHGs and anthropogenic perturbations is required to better constrain the controlling factors and mechanisms for GHG emissions from increasingly human-impacted river systems worldwide.

A small number of studies that measured simultaneously $CO_2$, $CH_4$, and $N_2O$ in large rivers such as the Amazon and Congo have revealed some common longitudinal patterns of gas concentrations determined by major sources and production

mechanisms (Richey et al., 1988; Borges et al., 2015). For instance, Richey et al. (1988) found large increases in dissolved $CO_2$ and $CH_4$ and slight reductions in $N_2O$ in the Amazon mainstem downstream of large floodplains. They suggested that predominant anaerobic metabolic processes might drive the enhanced release of $CO_2$ and $CH_4$ from floodplains, removing $N_2O$ concomitantly through more efficient denitrification under oxygen-depleted conditions. A regional-scale comparison of these three GHGs in 12 African rivers also showed similar relationships between the three gases and wetland extents within the

basin (Borges et al., 2015). These previous studies have used positive relationships between the concentrations of dissolved organic carbon (DOC) and GHGs as an indication of the riverine heterotrophy driving in-stream production of those GHGs (Borges and Abril, 2011; Borges et al., 2015; Teodoru et al., 2015). The widely used concept of river continuum from headwater to mouth assumes gradual and continual changes in riverine organic matter composition and metabolic processes corresponding to downstream variations in environmental conditions and biotic communities along the river (Vannote et al.,

1980). This concept of river continuum has been successful in describing longitudinal patterns of the quantity and lability of dissolved organic matter (DOM) (Koehler et al., 2012; Catalán et al., 2016). However, it remains unexplored whether the traditional view of gradual longitudinal variations in riverine organic matter composition and its transformations can explain riverine dynamics of different GHGs driven by a combination of multiple environmental factors specific to various natural and human-impacted river systems.

Several recent studies conducted in highly human-impacted river systems have found unique longitudinal and seasonal patterns of $CO_2$ and other GHGs that might be explained by different factors and mechanisms from those relevant to large pristine rivers (Silvennoinen et al., 2008; Crawford et al., 2016; Smith et al., 2017; Borges et al., 2018). A growing number of dams constructed on rivers worldwide have altered riverine flows of water and materials to the oceans, trapping over 100 billion metric tons of sediment and up to 3 billion metric tons of C in the reservoirs constructed over the last five decades (Syvitski et

al., 2005). The trapping of sediments and nutrients in reservoirs, combined with increased water retention time and improved light conditions promoting primary production, can alter significantly the rate of production and consumption of $CO_2$ and $CH_4$ in impounded waters (Maavara et al., 2017). Many studies of impoundment effects on GHG emissions from various types and sizes of dams including hydroelectric dams have reported contrasting results such as large emissions of $CO_2$ and $CH_4$ from the flooded vegetation and sediments following dam construction and an enhanced primary production and $CO_2$ sink in eutrophic

reservoirs (Abril et al., 2005; Chen et al., 2009; Barros et al., 2011; Hu and Cheng, 2013; Maeck et al., 2013; Crawford et al., 2016; Maavara et al., 2017; Shi et al., 2017). GHG dynamics in impounded waters and sediments may be explained by temporal changes in a suite of concomitant metabolic processes including primary production, methanogenesis, methane oxidation, nitrification, and denitrification (Beaulieu et al., 2010; Crawford et al., 2016; Park et al., 2018). However, little is known about complex interplays among multiple factors and mechanisms concurrently affecting the production and consumption of $CO_2$

and other major GHGs. Enhanced primary production and anaerobic metabolism may play a determining role in the long-term changes in $CO_2$ and $CH_4$ emissions from the impounded waters and sediments (Wang et al., 2017b). External inputs of organic matter and nutrients have been suggested as a primary control on GHG emissions from urban streams and rivers (Alshboul et al., 2016; Smith et al., 2017). Longitudinal surveys conducted along urbanized rivers have observed large pulsatile increases

in $CO_2$ and other GHGs in rivers receiving the effluents from wastewater treatment plants (WWTPs) (Garnier et al., 2013;
Burgos et al., 2015; Alshboul et al., 2016; Yoon et al., 2017). WWTP effluents and other anthropogenic sources can not only influence directly riverine GHG dynamics via discharged loads of GHGs, DOC, and nutrients but also modify riverine metabolic regimes (Garnier and Billen, 2007; Park et al., 2018). The complexity of involved metabolic processes, together with the multiplicity of environmental conditions and anthropogenic perturbations, poses a great challenge for a systematic understanding of GHG dynamics in human-impacted river systems.

The Han River basin exhibits unique longitudinal variations in dominant land use and anthropogenic perturbations, offering a venue for various biogeochemical studies of riverine organic matter and $CO_2$ dynamics (Fig. 1; Jung et al., 2012; Jin et al., 2016; Yoon et al., 2017). While headwater streams feeding into the upper section of the North Han River (hereafter termed "upper reach") drain largely forested mountainous terrain, the recent expansion of croplands on steep slopes has increased the release of nutrients and suspended sediment to downstream waters receiving agricultural runoff, particularly during heavy
monsoon rainfalls (Park et al., 2010). A series of dams on the middle section of the North Han River ("middle reach") may create "serial discontinuity" in riverine metabolic processes and GHG emissions (Ward and Stanford, 1983; Park et al., 2018), while urban tributaries draining the Seoul metropolitan area with a population >25 million discharge pulsatile loads of nutrients, DOC, and GHGs to the lower Han River ("lower reach") (Yoon et al., 2017). Building on our previous studies focused on $CO_2$ dynamics along the Han River (Yoon et al., 2016, 2017), this study aims to compare basin-wide patterns and controls of surface
water $CO_2$, $CH_4$, and $N_2O$. The primary objective was to examine the effects of dams and urban wastewater on the longitudinal variations in the concentrations of the three GHGs. Another important goal was to track changes in major sources of the three gases and DOM along the three river reaches by combining C isotope ratios in DOM, $CO_2$, and $CH_4$ with a basin-scale survey of gas concentrations and ancillary water quality parameters including DOM optical properties. Unlike other studies focused on the quantitative importance of inland waters as sources of GHGs, this study aims to examine basin-scale patterns of riverine
GHG dynamics against the conceptual frameworks that can explain longitudinal variations in riverine GHGs and DOM (refer to Park et al., 2018 and references therein), so only concentration data are presented without providing estimated emissions of the three GHGs. The comparison of the three GHGs, DOC, and other ancillary water quality measurements across the three reaches affected by different anthropogenic perturbations would provide empirical data that can be incorporated into the emerging concept of anthropogenic discontinuities in riverine metabolic processes involved in primary production, organic
matter degradation, and GHG emissions.

## 2 Materials and methods

### 2.1 Study site

The Han River (494 km), consisting of the North Han and South Han branches and the lower Han River (often called "Han River"), drains an area of 35,770 $km^2$ in the middle of the Korean Peninsula (Fig. 1; Seoul Metropolitan Government, 2017).
The mean annual precipitation in the Han River basin was 1323 mm for the period from 1983 to 2014, with up to 70% of the

annual precipitation concentrated during four months from June to September. Major land uses in the basin include forests (73.6%), croplands (14.1%), urban and industrial areas (2.6%), and other uses (9.7%) (Fig. S1). The highly urbanized metropolitan area along the lower reach has a large impermeable surface regarded as urban land use, accounting for 58% of the total city area of Seoul (Seoul Metropolitan Government, 2017). Sampling sites were established along a North Han River

section and the lower Han River, covering forested headwater (HR1) and agricultural streams along the upper reach through the middle and lower reaches impounded by dams and submerged weirs to the estuary.

Along the monitored reaches of the North Han River, four large multi-purpose hydroelectric dams have been constructed since 1943. The most upstream dam (Soyang; HR5) constructed in 1973 is the largest (height: 123 m; reservoir surface area: 70 km$^2$). The most downstream dam constructed in 1973 (Paldang; HR10) located at the confluence of the North and South Han

branches regulates river flow to the lower reach and supplies 2.6 million metric tons of water per day to the 25 million metropolitan population. Three major urban tributaries [Joongnang River (JN; HR12), Tan River (TC), and Anyang River (AY)] feed into the channelled lower Han River with submerged weirs at both ends. The tributary JN drains an area of 300 km$^2$ inhabited by 3.6 million, ~45% of which urban land use accounted for in 2014 (Seoul Metropolitan Government, 2017). Three WWTPs release treated sewage at the rate of 1.5 million m$^3$ d$^{-1}$ to the tributary (JN), with the bulk (1.3 million m$^3$ d$^{-1}$)

discharged from the monitored WWTP near the mouth of JN, which employs tertiary treatments including modified Ludzack Ettinger (MLE) and anaerobic-anoxic/oxic process (A$_2$O) (Ministry of Environment, 2015). Four WWTPs located within Seoul release effluents from tertiary treatments at the rate of 4.3 million m$^3$ d$^{-1}$.

## 2.2 Sampling and field measurements

Basin-wide surveys were carried out at 15 sites encompassing major land uses and human-induced perturbations in the Han

River basin during a summer monsoon period (July 2014) and in a dry period (May 2015) (Table 1; Figs. 1, S1). Compared to the upper reach (HR1–HR4) located in a heavily forested watershed with some scattered agricultural areas (Fig. S1), the impounded middle reach (HR5–HR11) and the lower reach receiving heavy loads of urban sewage (HR12–HR15) are subject to stronger anthropogenic perturbations. At 6 selected sites (HR1, HR2, HR4, HR8, HR11, and HR14), routine monitoring was continued at monthly intervals from July 2014 to July 2015, except for two winter months (January and February 2015).

In addition to two-season surveys, 13 samplings were conducted at an outlet of the tributary JN (HR12) from May 2015 to December 2017. In November 2015 and May 2016, additional samples were collected at 8 locations from a forested headwater stream to the outlet along the JN (Fig. 1; Fig. S1). The 8 sites were selected to cover the spatial pattern of land use, ranging from the forested upper reach to the increasingly urbanized downstream reaches (Fig. S1). The tributary samplings were complemented with 4 samplings at a discharge from a WWTP located a few km upstream of the JN outlet. To explore spatially

resolved patterns of riverine concentrations of three GHGs along the lower Han River, water and gas samples were collected at 8 locations during a boat expedition on 10 June 2016 that also aimed at continuous underway measurements of $p$CO$_2$ using automated $p$CO$_2$ measurement systems as described by Yoon et al. (2017).

Water samples were collected from a depth of 10–20 cm below the water surface in the middle of the stream channel or > 2 m away from the river bank. Spot water samples for chemical analysis were collected through a peristaltic pump into acid-washed amber glass bottles. At the same sampling point we measured water temperature, pH, electrical conductivity (EC), and dissolved oxygen (DO) using a portable pH meter (Orion 5-Star Portable, Thermo Scientific, USA). Surface water concentrations of three GHGs were measured by a manual headspace equilibration method (Kling et al., 1992; Yoon et al., 2016). Headspace equilibration was performed using a 60 ml polypropylene syringe to collect a 30 mL water sample and then a 30 mL ambient air sample. Another air sample was collected to measure gas concentrations in the ambient air. The syringe containing the water and air samples was shaken vigorously for 2 min; then ~20 mL of the equilibrated air was stored in a pre-evacuated 12 mL Exetainer vial. The stored gas sample volume was larger than the vial volume to create overpressure and hence minimize gas concentration change associated with potential gas leakage. Vials had been flushed with high-purity $N_2$ gas before the filled $N_2$ gas was evacuated by a pump. Gas analysis was performed usually within three days after the sampling. Air temperature and barometric pressure were measured in situ by a portable sensor. Barometric pressure, together with water temperature, was used to calculate concentrations of three GHGs from the gas concentrations of the equilibrated air and ambient air samples based on Henry's law (Hudson, 2004). Details on the calculation algorithms and Henry's law constants for three GHGs are provided in Hudson (2004).

## 2.3 Laboratory analyses

The equilibrated headspace air sample, as well as a sample from the ambient air used for equilibration, was injected into a GC (7890A, Agilent, USA) equipped with an FID coupled with a methanizer (for analysis of $CH_4$ and $CO_2$), a µECD (for $N_2O$ analysis), and a Supelco Hayesep Q 12 ft 1/8" column for the simultaneous measurement of three GHGs. High-purity $N_2$ gas (99.999%) was used as carrier gas at a constant flow rate of 25 mL min$^{-1}$. The flow rate of the reference gas was 5 mL min$^{-1}$ for FID ($N_2$) and 2 mL min$^{-1}$ for µECD (Ar/$CH_4$).Temperature settings include the inlet at 250°C, the oven at 60°C, the valve box at 100°C, FID at 250°C, and µECD at 300°C. The volume of the sample loop was 1 mL. Standard reference gases of three GHGs in $N_2$ balance (RIGAS Corporation, Korea) were used to calibrate the GC signals. The gas concentrations in these calibration standards were verified in the gas analysis laboratory of Korea Research Institute of Standards and Science (KRISS) that established the national measurement standards of gravimetrically prepared $CO_2$ in dry air and certified by a suite of gas analysis (Min et al., 2009). Additional headspace equilibration samples collected during the boat cruise in June 2016 were analysed for stable C isotope ratios of $CO_2$ ($\delta^{13}C_{CO2}$) and $CH_4$ ($\delta^{13}C_{CH4}$) by a GasBench-IRMS (ThermoScientific, Bremen, Germany) at the UC Davis Stable Isotope Facility.

Water samples were filtered through pre-combusted (450°C) glass fiber filters (GF/F, Whatman; nominal pore size 0.7 µm) at the laboratory. The concentration of total suspended solid (TSS) was determined by filtering a known volume of water sample through a pre-weighed GF/F filter and then weighing the filter again after drying at 60°C for 48 hours. Filtered water samples were analyzed for dissolved organic C, fluorescence excitation-emission matrices (EEMs), UV absorbance, total alkalinity

(TA), dissolved ions, and chlorophyll *a* (Chl *a*). For quality control of all chemical analyses, standards with known concentrations and ultrapure water were analyzed for every batch of ten samples and triplicate analysis was performed for approximately 10% of all analyzed samples to assess instrumental stability and accuracy.

DOC was measured by a total organic carbon (TOC) analyzer using high-temperature combustion of OM followed by thermal detection of $CO_2$ (TOC-$V_{CPH}$, Shimadzu, Japan). UV absorbance was measured across the wavelength range from 200 to 1100 nm using a UV-Vis spectrophotometer (8453, Agilent, USA). Fluorescence EEMs were collected on a fluorescence spectrophotometer (F7000, Hitachi, Japan) by simultaneous scanning over excitation wavelengths from 200 to 400 nm at 5 nm interval and emission wavelengths from 290 to 540 nm at 1 nm interval. Scan speed was 2400 nm min$^{-1}$ and the bandwidth was set to 5 nm for both excitation and emission. UV absorbance and fluorescence data were used to calculate specific UV absorbance at 254 nm (SUVA$_{254}$) (Weishaar et al., 2003), fluorescence index (FI) (McKnight et al., 2001), humification index (HIX) (Zsolnay et al., 1999), and common fluorescence EEM components (Fellman et al. 2010). Three fluorescent EEM components termed C1 (excitation/emission peaks at 325/467 nm), C2 (315/404 nm), and C3 (275/354 nm) were used to represent humic-like, microbial humic-like, and protein-like fluorescence, respectively (Fellman et al., 2010). TA was determined with 40–80 mL filtered samples on an automated electric titrator (EasyPlus Titrator Easy pH, Metrohm, Switzerland) based on the Gran titration method (Gran, 1952). Strong acid (0.01 N HCl) was used to titrate well beyond the equivalence point at pH between 4 and 3. Ions dissolved in water samples were analysed on an ion chromatograph (883 Basic IC plus, Metrohm, Switzerland). Chl *a* was analyzed spectrophotometrically following filtration on GF/C filters and acetone extraction (American Public Health Association, 2005).

Dual carbon isotopes analyses were conducted for water samples collected from six monthly monitoring sites and the JN WWTP effluents. Some filtered water samples were kept frozen before analysis. For $\Delta^{14}$C-DOC analysis, each water sample was acidified by 40% $H_3PO_4$ solution and sparged by helium gas to remove inorganic carbon. Then, the sample was oxidized with ultrahigh purity oxygen gas using UV lamp for 4 hours (Raymond and Bauer, 2001). The oxidized $CO_2$ was separated cryogenically with liquid $N_2$ in a vacuum line and sealed in a pre-baked pyrex tube. The $CO_2$ samples in the sealed pyrex tubes were sent to the National Ocean Sciences Accelerator Mass Spectrometry (NOSAMS) facility at the Woods Hole Oceanographic Institution for dual isotope analysis ($\Delta^{14}$C and $\delta^{13}$C) where accelerated mass spectrometer and isotope ratio mass spectrometer were used for $\Delta^{14}$C and $\delta^{13}$C analysis, respectively (http://www.whoi.edu/nosams/home). The $CO_2$ samples were transformed to graphite targets for $\Delta^{14}$C analysis by accelerated mass spectrometry.

**2.4 Statistical analysis**

Most dissolved GHGs and water quality data were not normally distributed, and their distribution patterns varied with river reach and measured parameter. Therefore, Kendall rank correlation, as a non-parametric test, was performed using the Kendall package of R (R Development Core Team, 2018) to investigate the relationships between water quality variables and dissolved GHGs both for the whole Han River basin and for each of the upper, middle, and lower reaches. For the monthly monitoring data, seasonal differences in three GHGs were analysed by the non-parametric Kruskal-Wallis analysis of variance followed

by Dunn's multiple comparisons. For reach-specific analysis, sampling sites were grouped into three reaches, as shown on Fig. 1. The upper reach includes sites HR1 to HR4 (outlet of the major tributary to the Lake Soyang, HR5). The middle reach covers the first dam on the monitored transect (HR5) through three downstream dams and ends up at HR 11,which receives discharge from the most downstream dam (HR10). The lower reach includes the mainstem (HR13, HR14, and HR15) and tributary sites [TC, JN (HR12), and AY] downstream of the first submerged weir in Seoul (Fig. 1). Relationships between three GHGs and water quality measurements were examined by regression analysis conducted with the data pooled for the whole basin and each cluster of three reaches and urban tributaries. Spatial variations among three reaches were further examined by principal component analysis (PCA).

## 3. Results

### 3.1 Longitudinal variations in GHG concentrations and DOM characteristics along the Han River

Two basin-wide surveys from the forested headwater stream (HR1) through the impounded middle reach to the highly urbanized lower reach, combined with the monthly monitoring at six selected sites, revealed distinct longitudinal patterns of $pCO_2$ and concentrations of $CH_4$ and $N_2O$ along the Han River (Figs. 2, 3; Tables 1, S1, S2, S3). The levels of the three measured GHGs were generally low along the upper reach, although noticeable increases occurred in the agricultural stream (HR2) compared with the very low values found in the forested headwater stream (HR1) (Table 1; Figs. 2, 3). The levels of $pCO_2$ in the middle reach ranged from 51 to 2465 μatm, with an overall mean value of 551 μatm for all measurements at 7 middle-reach sites (Tables 1, S1, S2, S3). The $pCO_2$ values at the four dam sites, ranging from 51–761 μatm, averaged 304 μatm, lower than the level expected for atmospheric equilibrium (~ 435 μatm). The mean $pCO_2$ was 1109 μatm in the upper reach (range: 163–2539 μatm) and 2587 μatm in the lower reach (range: 78–11298 μatm), both exceeding the average value of atmospheric equilibrium by a factor of 2.5 and 5.9, respectively. The levels of $pCO_2$ tended to be higher in summer than in other seasons at all monthly monitoring sites except HR8 and HR 11 located in the middle reach (Fig. S2).

Both $CH_4$ and $N_2O$ were always supersaturated with respect to atmospheric equilibrium ($CH_4$: 3.3 nmol $L^{-1}$; $N_2O$: 6.5 nmol $L^{-1}$) in all waters except the forested stream HR1 (Table 1; Figs. 2, 3). Compared with the very low concentrations measured at the forested stream (1.2–7.2 nmol $L^{-1}$), $CH_4$ concentrations were relatively high at the downstream agricultural site HR2 (41 – 1719 nmol $L^{-1}$) and three dam sites (HR6, HR7, and HR10; 693–748 nmol $L^{-1}$) (Table 1). $N_2O$ concentrations also increased from the low range of 8.9–20 nmol $L^{-1}$ in the forest stream to 28–46 nmol $L^{-1}$ in the agricultural stream and 212 nmol $L^{-1}$ at a dam (HR6). Similar to $pCO_2$, $CH_4$ and $N_2O$ exhibited very high values along the lower reach separated by two submerged weirs on both ends (HR12 – HR14). The concentrations of $CH_4$ and $N_2O$ at these four lower-reach sites averaged 1812 nmol $L^{-1}$ (112–6496 nmol $L^{-1}$) and 298 nmol $L^{-1}$ (22–1396 nmol $L^{-1}$), respectively. The concentrations of the three GHGs at some lower-reach sites were similar to or exceeded the levels found in the three tributaries draining the urban sub-catchments located in the Seoul metropolitan area (Fig. 2). There was no clear seasonality in $CH_4$ and $N_2O$ across the sites, although the

concentrations of these two gases at the lower-reach site HR14 tended to be higher in spring and summer than in fall and winter (Fig. S2).

The levels of DOC and FI exhibited overall downstream increases along the mainstem toward the river mouth with the exception of relatively high FI values found in the agricultural stream (Fig. 3; Tables S1, S2, S3). When Mann-Whitney $U$ tests were conducted to detect downstream changes between two successive sites, both DOC and FI were significantly different between the two mainstem sites (HR11, HR14) and the urban tributary JN (HR 12). Downstream increases in FI from the mean of 1.26 at the forest stream (HR1) to 1.50 at the most downstream site (HR15) may indicate an increasing contribution from the autochthonous DOM fraction (McKnight et al., 2001) Consistent with the increasing downstream trend of FI, HIX, as a measure of humification in soil organic matter (Zsolnay et al., 1999), decreased from the mean of 7.67 at the forest stream to 2.06 at HR 14 (the most downstream monthly monitoring site) and 0.79 at the estuarine site HR15, exhibiting significant changes over the transitions from HR1 to HR2 and from HR4 to HR8 (Fig. 3, Tables, S1, S2, S3). The concentrations of three measured nutrients ($NH_4^+$, $NO_3^-$, and $PO_4^{3-}$) were generally higher at the agricultural stream (HR2) and the lower-reach sites (HR12 and HR14) than at the other sites (Fig. 3; Tables S1, S2, S3). Given the relatively small proportion of tributary discharge in the mainstem flow ranging from ~5% in the monsoon period to 12% in dry seasons, the comparison of monthly water quality measurements between the six sites and the urban tributary (HR12) illustrates the disproportionate influence of urban tributary inputs on the downstream increases in the concentrations of DOC and nutrients observed in the lower reach (Fig. 3). The urban tributary generally exhibited high levels of DOC, FI, protein-like DOM fluorescence (C3/DOC), and the three major nutrients ($NH_4^+$, $NO_3^-$, and $PO_4^{3-}$), but relatively low values of pH, DO, HIX, and Chl $a$ (Fig. 3; Tables S1, S2, S3). Downstream changes in all water quality measurements from HR11 downstream of the last cascade dam to HR14 in the middle of the lower reach reflected either very high or very low values observed in the tributary (HR12) compared to the values observed in the impounded middle reach.

When we aggregated measurements as the three mainstem reaches and the group of the three urban tributaries, at least one of the three GHGs exhibited a significant negative relationship with pH ($pCO_2$) and DO ($pCO_2$ and $CH_4$), and a positive relationship with DOC ($CH_4$ and $N_2O$) (Fig. 4). The significant negative or positive relationships identified for each aggregated data set generally conformed to the overall trends shown for the whole basin (Fig. 4). A positive relationship between DO and $N_2O$ established in the lower reach was noticeable given no significant relationship found for the other reaches. Reach-specific clustering of data was also found on a PCA scatter plot with two primary components accounting for 60.3% of the total data variation (Fig. S3). While the middle-reach data on the PCA plot overlapped considerably with the large scatter shown by the upper reach, the majority of the lower-reach data and most tributary data points were separated from the regions covered by the upper and middle reaches.

Kendall rank correlation analyses conducted with all measurements grouped for each of three reaches revealed reach-specific patterns of significant correlations for the three GHGs (Fig. 5; Table S4). Significant negative correlations were found between $pCO_2$ and three water quality parameters – pH, DO, and Chl $a$ in the lower reach. These negative correlations were either absent for $N_2O$ or significant only with DO in the case of $CH_4$. Compared to no or weak correlations observed for the upper

and middle reaches, some parameters measured in the lower reach, including water temperature, $PO_4^{3-}$, C1/DOC, and C2/DOC, exhibited strong positive correlations with $pCO_2$. Values of $pCO_2$ measured at the middle reach sites had some negative correlations with pH, TA, water temperature, cations ($Na^+$, $K^+$, and $Mg^{2+}$), and anions ($Cl^-$ and $SO_4^{2-}$). These shifting correlations across the reaches were also found for $CH_4$ and $N_2O$, but to varying degrees. $CH_4$ in the lower reach had significant, but relatively weak correlations with $PO_4^{3-}$, C1/DOC, and C2/DOC, whereas $N_2O$ showed strong positive correlations with parameters such as EC, TA, all measured cations, $Cl^-$, $NO_3^-$, $SO_4^{2-}$, DOC, FI, and C3/DOC. In the case of $CH_4$, stronger correlations were found in the upper reach, including pH, EC, TA, cations ($K^+$, $Ca^{2+}$, and $Mg^{2+}$), anions ($Cl^-$ and $SO_4^{2-}$), FI, and C3/DOC, but these correlations were weaker or insignificant in the middle and lower reaches.

## 3.2 Longitudinal variations in GHG concentrations and DOM characteristics along an urban tributary

Two synoptic samplings along the urban tributary (JN), complemented with additional samplings at a tributary outlet (HR12) and an upstream WWTP discharge, revealed the dominant influence of WWTP effluents on the concentrations of the three GHGs, DOC, and nutrients measured at the tributary outlet (Fig. 6; Table S5). All the three GHGs exhibited similar levels and variations in the WWTP effluents and the tributary outlet, indicating a strong contribution of treated wastewater to the tributary gas export to the lower Han River. Both $pCO_2$ and $N_2O$ concentrations in the tributary abruptly increased along the terminal section downstream of the WWTP. In contrast, $CH_4$ concentrations were very low (4.7–10 nmol $L^{-1}$) at the forested headwater stream, exhibited large fluctuations along the middle reach (279–5838 nmol $L^{-1}$), and remained similar or decreased slightly ranging from 422 to 6496 nmol $L^{-1}$ in the terminal section downstream of the WWTP. The levels of the three gases displayed similar large temporal variations at the WWTP effluents and the tributary outlet (Fig. 6e, 6f, 6g). The observed temporal variations in gas concentrations were not significantly related to fluctuations in temperature, precipitation, and streamflow measured near the tributary outlet site, although the three gases tended to be lower in concentration during wet summer months in 2016 and 2017 (Fig. 6d).

## 3.3 C isotope ratios in DOM, $CO_2$, and $CH_4$

Despite large scatters on the dual plot of $\delta^{13}C$ and $\Delta^{14}C$ in DOM sampled across the Han River basin, the dual isotopic signatures for the urban tributary JN, WWTP effluents, and the most downstream site HR14 in the lower reach were clearly separated from the values found at the forested headwater stream HR1 (Fig. 7; Table S6). In both the wet-season (July 2014) and dry-season (May 2015) samplings, $\Delta^{14}C$ was relatively high at the forested stream HR1, with a higher value in July 2014 (58.9‰; modern age) than in May 2015 (−29.6‰; 180 years old). $\Delta^{14}C$ values exhibited an overall decreasing downstream trend toward the lower-reach site HR14 (July 2014: −78.2‰, 590 year old; May 2015: −87.9‰, 675 years old), with higher values observed across the mainstem sites in July 2014 (–129.1 to 66.8‰) than in May 2015 (–113.5 to –29.6‰). An exceptionally high $\Delta^{14}C$ value (66.8‰) was found in July 2014 at the second most downstream site (HR11) located 17 km downstream of the last of the cascade dams, indicating an influence of $^{14}C$-enriched DOM discharged from the upstream dams.

The values of $\Delta^{14}C$ measured in the urban tributary and WWTP effluent (around −100‰) were lower than those measured at
all the mainstem sites. $\delta^{13}C$ was highly variable at two middle-reach sites and increased almost linearly from the second most
downstream site through the urban tributary and WWTP effluents to the last site (Fig. 7).

The concentrations of the three GHGs and the values of $\delta^{13}C$ in $CO_2$ and $CH_4$ (Fig. 8; Tables S7, S8) measured along a cruise
transect exhibited large increases in gas concentrations and either gradual increases in $\delta^{13}C_{CO2}$ or abrupt decreases in $\delta^{13}C_{CH4}$
along the confluence of the urban tributary JN (HR12). $\delta^{13}C_{CO2}$ values ranged from −20.9 to −16.7‰ (Table S8), which were
distinctly higher than the range of $\delta^{13}C_{DOC}$ measured at the same sites (−28.2 to −20.6‰) (Table S6). The gradual
downstream increases in $\delta^{13}C_{CO2}$ along the mainstem transect reflected the tributary contributions to the mainstem isotopic
composition, because the values found in the two upstream tributaries (−18.2‰, −18.3‰) and a downstream tributary (−14.7‰)
were higher than the upstream mainstem values (Table S8). The downstream tributary had lower $pCO_2$, higher $\delta^{13}CO_2$, and
higher concentrations of $CH_4$ and Chl $a$ relative to the upstream tributaries (Fig. 8). The values of $\delta^{13}C_{CH4}$ showed an overall
increasing trend along the lower reach downstream of the urban tributary JN (Fig. 8; Table S8). While the tributary
concentrations of $N_2O$ were generally higher than those of the mainstem sites, Chl $a$ concentrations were lower at the two
upstream tributaries (TC and JN), but higher at the downstream tributary AY than the values observed at the adjacent mainstem
sites (Fig. 8c).

## 4 Discussion

### 4.1 Reach-specific patterns and controls of the three GHGs

Building on our previous report on $CO_2$ dynamics in the Han River basin (Yoon et al., 2016, 2017), this study provided a more
comprehensive view of longitudinal patterns in the three GHGs across the upper, middle, and lower reaches affected by
different types and magnitudes of anthropogenic perturbations (Figs. 2−4). The three gases exhibited large longitudinal
variations resulting from gas-specific increases or decreases at impoundment-affected sites and localized concentration peaks
along the lower reach downstream of polluted urban tributary inflows. Although $pCO_2$ at some impoundment-affected sites
was very low, approaching or falling below the levels expected for atmospheric equilibrium, $CO_2$ and the other two gases were
generally supersaturated with respect to atmospheric equilibrium across the river basin (Table 1; Fig. 2). The very high levels
of $pCO_2$ observed in the mainstem (up to 4132 μatm) and the three urban tributaries(up to 11970 μatm) of the lower Han River
fall in the high ranges found in some polluted rivers in Europe (Kempe, 1984; Frankignoulle et al., 1998; Borges et al., 2006)
and China (Yao et al., 2007; Ran et al., 2015; Liu et al., 2016; Wang et al., 2017). Consistent with these previous reports on
$pCO_2$ and other studies reporting high levels of $CH_4$ and $N_2O$ in polluted urban waters (Garnier et al., 2013; Yu et al., 2013;
Smith et al., 2017; Wang et al., 2017b, 2018), the results observed in the lower Han River emphasize the dominant influence
of urban tributaries carrying wastewater as a primary anthropogenic source of GHGs in the highly urbanized river system.

Different longitudinal patterns of the three measured GHGs (Fig. 2), together with reach-specific significant correlations
between the GHGs and other measured water quality components (Fig. 5; Table S4), illustrate that the spatial distribution and

temporal dynamics of the three gases in the Han River basin may be significantly different from those found in pristine or less anthropogenically modified river systems. The extent of major natural sources such as wetlands and inundated floodplains has been considered as a primary factor for longitudinal variations in $CO_2$ and $CH_4$ in large rivers (Richey et al., 1988; Borges et al., 2013; Abril et al., 2014). Those natural sources are rarely found in the Han River basin, where the middle and lower reaches have been modified substantially by man-made structures. This lack of natural sources, combined with the differential patterns of the three GHGs attributed to dams and urban wastewater, suggests that increased water retention time and nutrient enrichment may play crucial roles in the production and consumption of the three GHGs in this highly regulated river system (Crawford et al., 2016). It would be very challenging to identify the primary drivers out of the multiple factors involved in the production and consumption of the three GHGs in urbanized river systems, because some factors responsible for explaining the mechanism for two gases (e.g., $CO_2$ and $CH_4$) may be totally invalid for the other gas (e.g., $N_2O$) (Smith et al., 2017; Wang et al., 2017b). However, the observed longitudinal patterns of the three GHGs (Figs. 2–4), along with their correlations with specific sets of water quality components (Fig. 5), make one thing clear. The primary factors and mechanisms for the production and consumption of these three gases may change in response to longitudinal variations in dominant anthropogenic perturbations, often abruptly as shown by the localized concentration peaks of GHGs downstream of urban tributary inflows (Figs. 2, 8).

The highly variable concentrations of $CH_4$ and $N_2O$ along the middle reach in contrast to the consistently low levels of $pCO_2$ at the dam sites (Figs. 2–4) suggest that the rates of concomitant metabolic processes involved in the production and consumption of these gases in reservoir water and sediments may vary with predominant dam conditions such as water depth and sediment accumulation. The low values of $pCO_2$ measured at the impoundment-affected sites including site HR11 downstream of the last dam (HR10) indicate an enhanced planktonic $CO_2$ uptake, in agreement with the lowered $pCO_2$ levels as a consequence of increased primary productivity in some eutrophic impounded reaches of the Mississippi (Crawford et al., 2016), the Yangtze (Liu et al., 2016), and a Yellow River tributary (Ran et al., 2017). However, some previous studies have reported drastic increases in $CO_2$ and $CH_4$ emissions from the flooded vegetation and soils in the initial years following dam construction (Abril et al., 2005; Chen et al., 2009; Shi et al., 2017). Given the negative relationship established between reservoir age and emissions of $CO_2$ and $CH_4$ from a wide range of hydroelectric reservoirs (Barros et al., 2011), the initial pulse-like gas emissions from the newly constructed reservoirs might gradually decrease with increasing reservoir age. Several studies have reported significant negative relationships between Chl $a$ and $pCO_2$ in eutrophic impounded rivers (Crawford et al., 2016; Liu et al., 2016; Ran et al., 2017). However, in the middle reach of the Han River, $pCO_2$ was not correlated with Chl $a$, but had some significant positive correlations with DOM optical properties such as HIX, C1/DOC, and C2/DOC (Fig. 4). While HIX and C1/DOC indicate the degree of humification and the proportion of terrestrial DOM components, respectively, C2/DOC represents the proportion of "microbial humic components" in the bulk DOM (Fellman et al., 2010; Parr et al., 2015). The observed concentration range of Chl $a$ might be too narrow to detect any significant correlation with $pCO_2$. In contrast, the relatively high levels of $pCO_2$ concurred with strong optical intensities of terrestrial DOM components at some middle

reach sites that are less affected by impoundments (e.g., HR 7 and HR8), resulting in the significant correlations between the relatively wide ranges of $pCO_2$ and DOM optical properties.

Although a small number of measurements of $CH_4$ and $N_2O$ and their inconsistent spatial patterns observed along the impounded middle reach (Fig. 2; Tables S2, S3) require a cautious assessment of impoundment effects on the concentrations of the two gases, the higher values of $CH_4$ and $N_2O$ in some dams and outflows compared to the upstream levels indicate dam-specific conditions driving the production and consumption of $CH_4$ and $N_2O$ in reservoir water and sediments. Crawford et al (2016) observed a weak summer-time $CO_2$ sink due to enhanced photosynthesis but elevated concentrations and fluxes of $CH_4$ along the upper Mississippi River impounded by a series of low dams constructed for river navigation. They attributed the observed $CH_4$ supersaturation to anaerobic conditions in organic-rich sediments. Despite no correlation between $CH_4$ and DO levels in the middle reach (Fig. 5; Table S4), the fact that $CH_4$ levels were higher at the three shallower dam sites (HR6, HR7, and HR10) than at the most upstream dam (HR5) with a maximum depth of 110 m implies the balance between anaerobic $CH_4$ production in bottom sediment and aerobic and anaerobic $CH_4$ oxidation in the water column with a vertical gradient of $O_2$ availability as a driving force for the observed spatial variations in $CH_4$ concentrations (Roland et al., 2017). Shallow reservoirs and river inflows accumulating methanogenic sediments have been identified as $CH_4$ emission hot spots, in which relatively short retention time of sediment-derived $CH_4$ in aerobic water column can lead to reduced rates of $CH_4$ oxidation compared with deeper reservoirs (Maeck et al., 2013; Beaulieu et al., 2014). $CH_4$ concentrations in the middle reach exhibited a weak, but significant correlation with DOC concentrations (Fig. 5, Table S4). This correlation may indicate an active methanogenesis in anaerobic reservoir sediments that is often accompanied by increases in surface water DOC concentrations (Chen et al., 2009; Wang et al., 2017b). It is also possible that some local sources of organic wastes surrounding the reservoirs may have directly discharged wastewater rich in DOC and $CH_4$ (Bergier et al., 2014; Wang et al., 2017b). Higher $CH_4$ concentrations measured at warmer temperatures are also in agreement with the increased rates of $CH_4$ production observed during warm summer months in a shallow reservoir in Ohio, USA (Beaulieu et al., 2014). N enrichment, alone or in combination with anaerobic conditions favourable for denitrification, has been suggested as a key control on $N_2O$ production in impoundments, although strictly anaerobic conditions might result in a more complete denitrification to $N_2$, producing little $N_2O$ (Beaulieu et al., 2015; Wang et al., 2017b). The lack of clear impoundment effects on $N_2O$ concentrations except for one reservoir (HR6; Fig. 2) can be explained by little $N_2O$ production in the other reservoirs or the complex interplay between $N_2O$ production from nitrification and denitrification and $N_2O$ consumption under changing availability of $O_2$ (Beaulieu et al., 2015).

Large increases in GHG concentrations along the lower reach may be a combined result of the net in-stream production and direct inputs from WWTPs. Direct influences of wastewater-derived GHGs have been observed in urban rivers receiving WWTP effluents (Garnier et al., 2013; Yu et al., 2013; Burgos et al., 2015; Alshboul et al., 2016; Wang et al., 2017a; Yoon et al., 2017). For example, large pulsatile emissions of $CH_4$ and $N_2O$ in the Guadalete River estuary in Spain were found near the discharge from a WWTP, as a combined result of direct gas emissions from WWTP effluents and indirect effects on the production of $CH_4$ and $N_2O$ in the water channel and bottom sediments downstream (Burgos et al., 2015). Previously we used a mass balance approach based on three cruise underway measurements of $pCO_2$ and DOC and the estimated rates of $CO_2$

outgassing from the same reach of the lower Han River, and additional measurements of $p$CO$_2$ and DOC at two urban tributaries (TC and JN) to show that the two tributaries JN and TC delivering WWTP effluents accounted for up to 72% of the

425 CO$_2$ concentration measured at a downstream location of the lower reach (Yoon et al., 2017). When the rates of CO$_2$ production, consumption, and outgassing were estimated using the mass balance approach for a section upstream and two sections downstream of the two tributaries lower reach in June 2016 (Yoon et al., 2017), the amount of CO$_2$ produced from organic matter biodegradation was much greater than the amount of CO$_2$ consumed by phytoplankton and similar to the CO$_2$ outgassing to the atmosphere. In May 2015, when Chl $a$ concentrations were much higher than in June 2016, the mass balance suggested

that the bulk of CO$_2$ delivered by the tributaries might have been consumed by phytoplankton photosynthesis in the downstream section of the lower reach (Yoon et al., 2017). As observed in other polluted rivers enriched in labile DOM moieties derived from urban sewage (Guo et al., 2015), newly produced CO$_2$ from active microbial decomposition can sometimes exceed the phytoplankton uptake of CO$_2$ depending on the prevailing environmental conditions. By directly measuring $\delta^{13}$C in CO$_2$ respired by bacterioplankton in two streams and eight lakes in Canada, McCallister and del Giorgio

(2008) showed that the production of CO$_2$ through bacterial degradation of terrigenous DOM decreased in sharp contrast to the increasing proportion of algal-derived DOC and CO$_2$ with increasing levels of Chl $a$. Their findings implied an algal-driven activation of metabolism in eutrophic freshwater systems, although it would require further research to verify the findings in the boreal freshwaters in temperate and other biomes. It remains largely unexplored how the balance between autotrophy and heterotrophy in many eutrophic river systems found across the temperate zones shifts in response to changing environmental

conditions (Garnier and Billet, 2007).

Large down-river increases (Fig. 2) and temporal variability (Figs. 3, S1) in CH$_4$ and N$_2$O along the lower reach may result from a combination of processes including direct WWTP discharge and in-stream production and consumption. As indicated by the similar ranges of CH$_4$ and N$_2$O found in the WWTP effluents and the tributary JN outlet (Fig. 6), the amount of CH$_4$ and N$_2$O discharged from the WWTP appeared to drive the magnitude and temporal variability of the tributary inputs to the

445 lower reach. In the case of CH$_4$, however, the large spatial and temporal variations observed along the tributary upstream of the WWTP also indicate the potential role of bottom sediments as an upstream source of CH$_4$ (Stanley et al., 2016), although further research is needed to elucidate the in-stream production of CH$_4$ in the tributary. Despite the dominant influence of tributary exports, other riverine processes also need to be considered to explain the complex downstream spatial patterns of these gases, as indicated by the gas-specific sets of significant correlations with measured water quality components (Fig. 5).

In contrast to the nonsignificant correlation between CH$_4$ and DO in the impounded middle reach, CH$_4$ concentrations in the lower reach exhibited a significant negative correlation with (DO) as well as positive correlations with PO$_4^{3-}$, water temperature, C1/DOC, and C2/DOC. As observed in other urbanized river systems (Beaulieu et al., 2015; Smith et al., 2015; Wang et al., 2018), CH$_4$ correlated positively with PO$_4^{3-}$, but negatively with DO, implying that the nutrient enrichment often leading to severe phytoplankton blooms during warm summer months may create favourable conditions for anaerobic

methanogenesis in the lower reach that is almost impounded by the two submerged weirs. Out of many water quality components that were significantly correlated with N$_2$O (Fig. 5), NH$_4^+$ and NO$_3^-$ have been considered as two primary

predictors of riverine $N_2O$, particularly in urban streams and rivers influenced by sewage (Yu et al., 2013, He et al., 2017; Smith et al., 2017). The lack of any significant correlation between $N_2O$ and DO in the lower reach may indicate a potential $N_2O$ production from nitrification and incomplete denitrification under widely varying levels of DO (Fig. 5).

## 4.2 Tracking sources of DOM and GHGs using C isotope ratios

The dual C isotope signatures of DOM measured at the five monthly monitoring sites and the outlet and WWTP discharge of the urban tributary JN overlapped considerably with the reported ranges for soil organic matter (SOM) and phytoplankton biomass (Fig. 7) (Raymond and Bauer, 2001; Marwick et al., 2015 and references therein). When Marwick et al. (2015) compiled 695 $\Delta^{14}C_{DOC}$ data collected in a wide variety of rivers around the world, most data points fell in the range from $-100$ to $+200‰$ and 72% of the $\Delta^{14}C_{DOC}$ data indicated a modern age, contrasting with the much smaller proportion of "modern" POC (22% of 483 $\Delta^{14}C_{POC}$ measurements). The relatively large site-to-site variations in both $\delta^{13}C_{DOC}$ and $\Delta^{14}C_{DOC}$ make it very difficult to evaluate the relative contributions of allochthonous and autochthonous sources to the isotopic signatures of the bulk DOM. While $\delta^{13}C_{DOC}$ was highly variable along the upper to middle reaches from HR2 to HR11, the values of $\delta^{13}C_{DOC}$ at the most downstream site HR14 ($-20.6‰$ and $-23.2‰$) were distinctively higher than those measured at the forested headwater stream ($-28.2‰$ on both sampling dates) (Table S6). The DOM optical properties measured at HR14 were also significantly different from the high HIX and low FI values indicating the predominance of soil-derived DOM in the headwater stream (Fig. 3; Table S1). Taken together, the isotopic composition and optical properties of DOM in the lower reach may reflect the downstream addition of DOM components derived from anthropogenic sources such as WWTP effluents ($\delta^{13}C_{DOC}$ around $-26‰$; Table S6) or plankton biomass (note the wide range of the plankton $\delta^{13}C_{DOC}$.in Fig. 7).

The distinct seasonal differences in $\Delta^{14}C_{DOC}$ across the five mainstem sites illustrate that the age of DOM is generally younger during the monsoon period (July 2014; modern to 590 years B.P.) than in the dry season (May 2015; 180 to 675 years B.P.) (Fig. 7; Table S6). This seasonality might have resulted from an increased contribution of DOM components released from terrestrial sources during monsoon rainfalls. The enrichment of $^{14}C$ in DOM identified in other temperate river systems has been attributed to a larger contribution from $^{14}C$-enriched litter and soil organic matter and shorter residence times of DOM in systems with higher annual precipitation (Raymond and Bauer, 2001; Butman et al., 2015). The longitudinal increase in DOM age from a modern age to 180 years B.P. at the forested headwater stream to 590–675 years B.P. at the most downstream site may reflect a preferential degradation of young, labile components during riverine DOM transport (Raymond and Bauer, 2001), but also indicates a significant contribution of aged DOM derived from downstream anthropogenic sources; for example, the age of DOM measured at the outlet and WWTP effluents of the urban tributary JN ranged from 765 to 1050 years B.P. (Table S6). The values of $^{13}C_{DOC}$ ($-25.8‰$) and $\Delta^{14}C_{DOC}$ ($-97.9‰$ and $-113.5‰$) measured in the WWTP effluents (Table S6) were similar to those reported by other studies (Griffith et al., 2009; Butman et al., 2015). WWTP effluents have been shown to contain aged organic matter with characteristic C isotopic composition (Griffith et al., 2009; Griffith and Raymond, 2011; Butman et al., 2015). As suggested by Griffith and Raymond (2011), aged DOM derived from the WWTP effluents (765–905

years B.P.; Table S6) may contain labile materials, which, mixed with other labile components from in-stream sources such as phytoplankton, can fuel the riverine heterotrophy along the lower reach.

While $\delta^{13}C$ has usually been measured for the total DIC consisting of dissolved $CO_2$, $HCO_3^-$, and $CO_3^{2-}$ to track downstream changes in DIC sources along streams and rivers (Barth et al., 2003; Schulte et al., 2011; Zeng et al., 2011; Deirmendjian and Abril, 2018), our measurements of $\delta^{13}C_{CO2}$ and $\delta^{13}C_{CH4}$ (Fig. 8; Table S8), combined with the dual C isotope signatures of DOM (Fig. 7; Table S6), also provided some insights into the contributions of treated wastewater and enhanced outgassing and phytoplankton growth to the riverine isotopic composition of $CO_2$ and $CH_4$ in the eutrophic lower reach. The values of $\delta^{13}C_{CO2}$ increasing from −20.9‰ at 76 km from the river mouth to −16.7‰ at 50 km fall within the range of $\delta^{13}C$ measured for $CO_2$ dissolved in riverine and estuarine waters (−25 – −15‰) (Longinelli and Edmond, 1983; Maher et al., 2013). However, the values reported here are less negative than the ranges of $\delta^{13}C$ measured directly for $CO_2$ respired by bacteria consuming organic matter of terrestrial and algal origin in two streams and eight lakes in Canada (−32.5 – −28.4‰) (McCallister and del Giorgio, 2008). The comparison of the lower-reach $\delta^{13}C_{CO2}$ values observed >10 km downstream of the tributary JN (−17.4 to −16.7‰; Table S8) with the $\delta^{13}C_{DOM}$ measured at the forest stream HR1 (−28.2‰; Table S6) and the most downstream site HR 14 (−20.6 to −23.2‰; Table S6) suggests that the $\delta^{13}C_{CO2}$ in the lower reach does not directly result from the isotopic composition of $CO_2$ originating from the microbial degradation of soil-derived DOM or the mixture of DOM components in the lower reach. In an inter-regional comparison of $\delta^{13}C_{DIC}$ measured at 318 streams across Sweden, Campeau et al. (2017) estimated $\delta^{13}C_{CO2}$ based on carbonate-equilibrium fractionation and $^{13}C$ enrichment between the three DIC species (Zhang et al., 1995). The $\delta^{13}C_{CO2}$ values estimated by Campeau et al. (2017) exhibited a significant negative relationship with DOC concentrations, with a particularly low median value of $\delta^{13}C_{CO2}$ (−25.3‰) matching with a high median DOC concentration (28.7 mg L$^{-1}$) in headwater streams in a heavily forested region. The very low $\delta^{13}C_{CO2}$ values in the DOC-rich forest streams in Sweden (Campeau et al., 2017) were close to the $\delta^{13}C$ values for $CO_2$ released from bacterial DOM degradation (McCallister and del Giorgio, 2008), but much lower than those observed along the lower Han River. Therefore, we need to consider other sources than the soil-derived DOM to account for the isotopic composition of $CO_2$ dissolved in the lower reach. Although we did not measure $\delta^{13}C_{CO2}$ and $\delta^{13}C_{CH4}$ in WWTP effluents, the comparison of the ranges of $\delta^{13}C_{CO2}$ and $\delta^{13}C_{CH4}$ observed in the lower reach and the three urban tributaries delivering large loads of treated wastewater (Fig. 8; Table S8) suggests that WWTP-derived gases can directly influence the isotopic composition of the two gases dissolved in the mainstem reach downstream of the tributary inflows.

We can also benefit from rich literature information on $\delta^{13}C_{DIC}$ to further constrain the observed 4‰ downstream increase in $\delta^{13}C_{CO2}$ along the lower reach (Fig. 8). Biogenic DIC originating from autotrophic respiration or mineralization of soil organic matter derived from C3 plants has a typical value around −27‰ but this $\delta^{13}C_{DIC}$ value can increase up to −23‰ in soil solution as a result of 1−4‰ isotopic enrichment caused by dissolution and gas exchange between the soil and atmosphere (Doctor et al., 2008; Schulte et al., 2011). When this biogenic DIC mixes with atmospherically equilibrated DIC and geogenic DIC dissolved from carbonates, both of which have a typical $\delta^{13}C$ value around 0‰, the resulting riverine $\delta^{13}C_{DIC}$ usually ranges from −15 to −5‰ (Telmer and Veizer et al., 1999; Barth et al., 2003; Schulte et al., 2011; Zeng et al., 2011). The values of

$\delta^{13}C_{DIC}$ estimated from our measurements of $\delta^{13}C_{CO2}$ based on carbonate equilibrium fractionation and the enrichment factors between the DIC species determined by Zhang et al. (1985) ranged from −13.0 to −8.8‰, reflecting the influence of both

biogenic and geogenic DIC. As mentioned earlier, the urban tributaries enriched in $\delta^{13}C_{CO2}$ relative to the mainstem might have contributed to the 4‰ downstream increase in the estimated $\delta^{13}C_{DIC}$. In addition, the enrichment may also reflect the net effect of a combination of concomitant processes including gas evasion to the atmosphere (Doctor et al., 2008) and photosynthesis (Finlay et al., 2004) that enrich the remaining DIC in $^{13}C$ through the preferential removal of the lighter $^{12}CO_2$. The outgassing of $CO_2$ can result in a pronounced increase in $\delta^{13}C_{DIC}$, such as the 3−5‰ enrichment along a relatively short

(0.5 km) transect of a forested headwater stream reported by Doctor et al. (2008). However, given the concomitant downstream increases in $\delta^{13}C_{CO2}$ and Chl $a$ concentrations, we cannot rule out a potential contribution of enhanced primary production in the eutrophic lower reach to the observed downstream$^{13}C$ enrichment. The high values of $\delta^{13}C$ observed in the tributaries, especially at the most downstream tributary AY showing the highest levels of $\delta^{13}C_{CO2}$ and Chl $a$, may also reflect that in-stream processes such as photosynthesis and $CO_2$ outgassing can modify the isotopic ratios of wastewater-derived $CO_2$,

resulting in the observed inter-tributary variations in $pCO_2$ and $\delta^{13}C_{CO2}$. Another potential source of isotopic enrichment might be the production of $CH_4$ from the reduction of $CO_2$ in the deeper water and bottom sediment, which consumes lighter C resulting in an enrichment of the remaining DIC pool (Barth et al., 2003). However, this mechanism might be less important than other described processes, because the longitudinal increase in $\delta^{13}C$ in $CH_4$ (Fig. 8) indicates an active oxidation occurring through the shallow downstream reach.

While agricultural activities, including rice cultivation, animal husbandry, and N fertilization, represent the primary anthropogenic source of $CH_4$ and $N_2O$ in anthropogenically impacted river systems (Silvennoinen et al., 2008; Garnier et al., 2013; Borges et al., 2018), the relative contribution of wastewater and landfills often increases drastically in urbanized watersheds (Yu et al., 2013; Smith et al., 2017; Wang et al., 2017b). Downstream changes in $CH_4$ concentrations along the lower reach reflect localized impacts of urban tributary inputs, which caused a large increase in the mainstem $CH_4$

concentrations from 136 to 3088 nmol $L^{-1}$ (Table S7) and a pronounced decrease in $\delta^{13}C_{CH4}$ from −36.6 to −48.6‰ (Table S8) across the inflows of the two tributaries TC and JN (Fig. 8). The concentrations of $CH_4$ peaked in the tributary inflow, but gradually decreased along the downstream reach toward the river mouth, indicating an efficient removal of $CH_4$ through oxidation and/or evasion to the atmosphere (Maher et al., 2013; Sawakuchi et al., 2016). Similar localized effects of WWTP effluents were observed in the lower reaches of Seine River, downstream of Paris and Rouen (Garnier et al., 2013). The

contrasting down-river trends of decreasing $CH_4$ concentrations and $^{13}C$ enrichment are consistent with the longitudinal patterns of $CH_4$ concentration and its stable C isotope ratios measured simultaneously in large rivers such as the Amazon River (Sawakuchi et al., 2016) and estuaries (Maher et al., 2013). Sawakuchi et al. (2016) found the increases in $\delta^{13}C_{CH4}$ and the abundance of a genetic marker for methane-oxidizing bacteria (*pmoA*) in waters with lower $CH_4$ concentrations across the mainstem and tributaries of the Amazon. They used stable isotopic mass balances of $CH_4$ in the water column and estimated

that 17–100% of $CH_4$ produced in the riverbed sediment may be oxidized during transport through water column to the atmosphere. During cruise expeditions employing a cavity ring-down spectroscope (CRDS) along a 15 km reach of the North

Creek estuary in Australia, Maher et al. (2013) observed increasing $\delta^{13}C$ values from $-61.07$ to $-48.62$‰ in contrast to large decreases in $CH_4$ concentrations from 74 to 2 nmol in the downstream direction. $CH_4$ oxidation was suggested as the primary driver of downstream increases in $\delta^{13}C$ in the studied estuary with relatively low levels of anthropogenic pollution (Maher et al., 2013). The down-river patterns of $CH_4$ concentration and isotopic composition observed in this study also suggest that $CH_4$ oxidation in the well-mixed, shallow water, in combination with physical evasion to the atmosphere, may efficiently remove $CH_4$ derived from multiple sources including the urban tributaries enriched in $CH_4$ and riverbed sediments affected by the eutrophic water and frequent phytoplankton blooms.

## 5 Implications for integrative concepts and future research

The three GHGs and ancillary water quality parameters measured in this study exhibited large basin-wide variations and localized peaks or reductions in the river sections affected by impoundments and urban wastewater. The observed longitudinal discontinuities deviating from the gradual downstream patterns of GHGs cannot be adequately explained by the traditional view of river continuum assuming gradual downstream changes in riverine metabolism and organic matter composition (Vannote et al., 1980). Although the river continuum concept has been useful in explaining gradual longitudinal variations in $CO_2$ from headwater streams to lowland rivers corresponding to the changing balance between autotrophy and heterotrophy (Koehler et al., 2012; Catalán et al., 2016; Hotchkiss et al., 2015), it has been increasingly recognized that rivers are often divided into discrete segments such as impoundments (Ward and Stanford, 1983; Poole, 2002) and eutrophic waters polluted by wastewater (Garnier and Billen, 2007; Yoon et al., 2017; Park et al., 2018). Previous studies of DOM biodegradation have often assumed a preferential degradation of labile components of riverine organic matter along the hypothetical continuum with minimal to low levels of anthropogenic perturbations (Koehler et al., 2012; Weyhenmeyer et al., 2012; Catalán et al., 2016). Our results emphasize the role of dams and wastewater as anthropogenic perturbations to the hypothetical continuum of DOM composition and lability envisioned in previous studies. For example, the significance levels for the correlations between the three GHGs and DOC or its optical properties varied along the three compared reaches of the Han River (Fig. 5; Table S4). This suggests that the DOM pool fuelling riverine heterotrophy may be continuously replenished by organic materials that are transformed from allochthonous components, or newly produced in the impounded middle reach and the eutrophic lower reach, or added as anthropogenic DOM components derived from agricultural runoff and urban wastewater. This enhanced heterotrophy, along with direct discharges of GHGs from WWTPs, may result in highly localized emission peaks of GHGs along the lower Han River and other eutrophic urban rivers worldwide.

The estimated rates for the production (respiration) and consumption (photosynthesis) of $CO_2$ in the lower reach downstream of major tributary inflows varied substantially depending on hydroclimatic conditions and plankton growth, indicating a shifting balance between autotrophy and heterotrophy driving $CO_2$ dynamics (Yoon et al., 2017). The different significance levels established between the three GHGs and measured nutrients ($NH_4^+$, $NO_3^-$, and $PO_4^{3-}$) implied some different roles that those nutrients may play in the production of each gas in the eutrophic lower reach (Fig. 5). The increased availability of P,

together with low DO levels and strong intensities of a DOM optical property indicative of "microbial humic-like" DOM (C2/DOC), was associated with the high levels of $pCO_2$ and $CH_4$. These correlations suggest that enhanced phytoplankton growth and anaerobic metabolic activity in the eutrophic reach often plagued by phytoplankton blooms may result in a net positive effect on the production of both $CO_2$ and $CH_4$ despite the immediate negative effect of plankton uptake of $CO_2$ on the surface water level of dissolved $CO_2$. In contrast, the levels of $NH_4^+$ and $NO_3^-$, rather than $PO_4^{3-}$, were more significantly correlated with $N_2O$ concentrations in the lower reach, in agreement with the well-established relationship between N levels and $N_2O$ production mechanisms in eutrophic rivers affected by urban sewage (Beaulieu et al. 2010; Yu et al., 2013; He et al., 2017). Considering the complexity of the multiple environmental factors and metabolic processes involved in the production and consumption of the three GHGs, future research needs to move beyond the simple correlation approach and directly measure rates of specific metabolic processes (e.g., $CH_4$ oxidation) and related biochemical or genetic markers (e.g., *pmoA*) to elucidate altered metabolisms and GHG emissions in anthropogenic river systems.

An important remaining question about the altered metabolic regimes in the anthropogenically impacted river system is how to evaluate the relative contributions of autochthonous gas production and external supplies of GHGs derived from WWTP effluents. In the context of anthropogenic river discontinuum (Park et al., 2018), impoundments and urban water pollution might be coupled through hydrologic connection and hence cause synergistic effects on downstream metabolic processes. In the case of the Han River basin, old cascade dams on the North Han River and large weirs newly constructed on the South Han River might be altering not only GHG dynamics but also DOM composition and lability. It warrants further research to explore how impoundment-induced changes in water retention and C biogeochemistry cascade down to affect organic matter transformations and GHG emissions in the eutrophic lower reach. A better understanding of altered metabolic processes and GHG dynamics in highly human-impacted river systems would contribute to establishing novel river basin management options integrating the traditional focus on water quality control and an emerging challenge of climate change mitigation by helping watershed managers set priority areas of policy responses to multiple environmental stresses.

**Data availability**

More data are available in supplementary information and can be requested from the corresponding author (jhp@ewha.ac.kr).

**Author contribution**

All authors contributed to data acquisition, discussion, and manuscript preparation. Manuscript writing was coordinated by J.-H. Park with contributions from all authors.

## Competing interests

The authors declare that they have no conflict of interest.

## Acknowledgements

This work was supported by the National Foundation of Korea (2014R1A2A2A01006577; 2017R1D1A1B06035179), "Cooperative Research Program for Agriculture Science & Technology Development (PJ012489022018)" by Rural Development Administration, Republic of Korea, and "Development of Gas Analysis Measurement Standards (18011051)" by Korea Research Institute of Standards and Science. We thank Borami Park for her assistance with sampling and analysis. We also thank Dr. Gwenaël Abril and three anonymous reviewers for their constructive comments.

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

**Table 1.** Summary of sampling sites and the concentrations of three GHGs measured during two basin-wide surveys at 15 sites and monthly monitoring at 6 sites in the Han River basin from July 2014 to July 2015. Monthly data are presented as means followed by ranges in parentheses. Note that additional data obtained at an urban tributary (HR 12) from May 2015 to December 2017 (N=13) are also presented as monthly data.

| Reach | Site | Name | Description | Coordinates | Distance to mouth (km) | $pCO_2$ (µatm) Monthly | July 2014 | May 2015 | $CH_4$ (nmol L$^{-1}$) Monthly | May 2015 | $N_2O$ (nmol L$^{-1}$) Monthly | May 2015 |
|---|---|---|---|---|---|---|---|---|---|---|---|---|
| Upper | HR1 | Haean | Forest stream | 38°15'N, 128°7'E | 299 | 1083 (515-2539) | 632 | 595 | 4.9 (1.2-7.2) | 1.2 | 13.3 (8.9-20) | 10.1 |
| | HR2 | Mandae | Agricultural stream | 38°16'N, 128°9'E | 292 | 1415 (530-2302) | 1498 | 530 | 529.0 (41-1719) | 287.8 | 36.4 (28-46) | 31.4 |
| | HR3 | Inbuk River | Tributary to Soyang | 38°6'N, 128°11E | 255 | | 606 | 163 | | 996.3 | | 11.8 |
| | HR4 | Soyang River | Inflow to reservoir | 38°0'N, 128°6'E | 234 | 960 (654-1541) | 1541 | 654 | 283.5 (104-967) | 104.1 | 12.0 (7.5-20) | 9.6 |
| Middle | HR5 | Soyang Dam | Reservoir | 37°56'N, 127°49'E | 189 | | 98 | 430 | | 59.0 | | 14.2 |
| | HR6 | Eoam Dam | Reservoir | 37°52'N, 127°41'E | 172 | | 246 | 128 | | 693.4 | | 212.2 |
| | HR7 | Cheongpyeong | Reservoir outflow | 37°43'N, 127°24'E | 122 | | 761 | 550 | | 738.0 | | 36.4 |
| | HR8 | North Han | N. Han outlet | 37°36'N, 127°20'E | 108 | 682 (56-1970) | 56 | 677 | 263.5 (71-472) | 378.3 | 17.1 (11-23) | 23.4 |
| | HR9 | South Han | S. Han outlet | 37°31'N, 127°22'E | 105 | | 73 | 173 | | 591.7 | | 18.8 |
| | HR10 | Paldang Dam | Reservoir | 37°30'N, 127°18'E | 97 | | 51 | 171 | | 747.8 | | 16.0 |
| | HR11 | Amsa | Lower Han River | 37°33'N, 127°7'E | 76 | 649 (72-2465) | 224 | 257 | 565.8 (210-870) | 595.5 | 21.1 (13-31) | 20.8 |
| Lower | HR12 | Joongnang R | Urban tributary | 37°33'N, 127°2'E | 66 | 8267 (4703-11970) | 7734 | 9369 | 2326.6 (422-6496) | 4112.4 | 429.8 (94-1396) | 1396.1 |
| | HR13 | Jamwon | Lower Han River | 37°31'N, 127°1'E | 63 | | 3179 | 1653 | | 2552.3 | | 140.2 |
| | HR14 | Bamseom | Lower Han River | 37°32'N, 126°55'E | 53 | 2356 (489-4132) | 3344 | 1463 | 1370.2 (265-2540) | 2033.8 | 69.3 (22-174) | 173.7 |
| | HR15 | Jeonryuri | Estuary | 37°41'N, 126°39'E | 23 | | 3544 | 2379 | | 111.5 | | 88.1 |

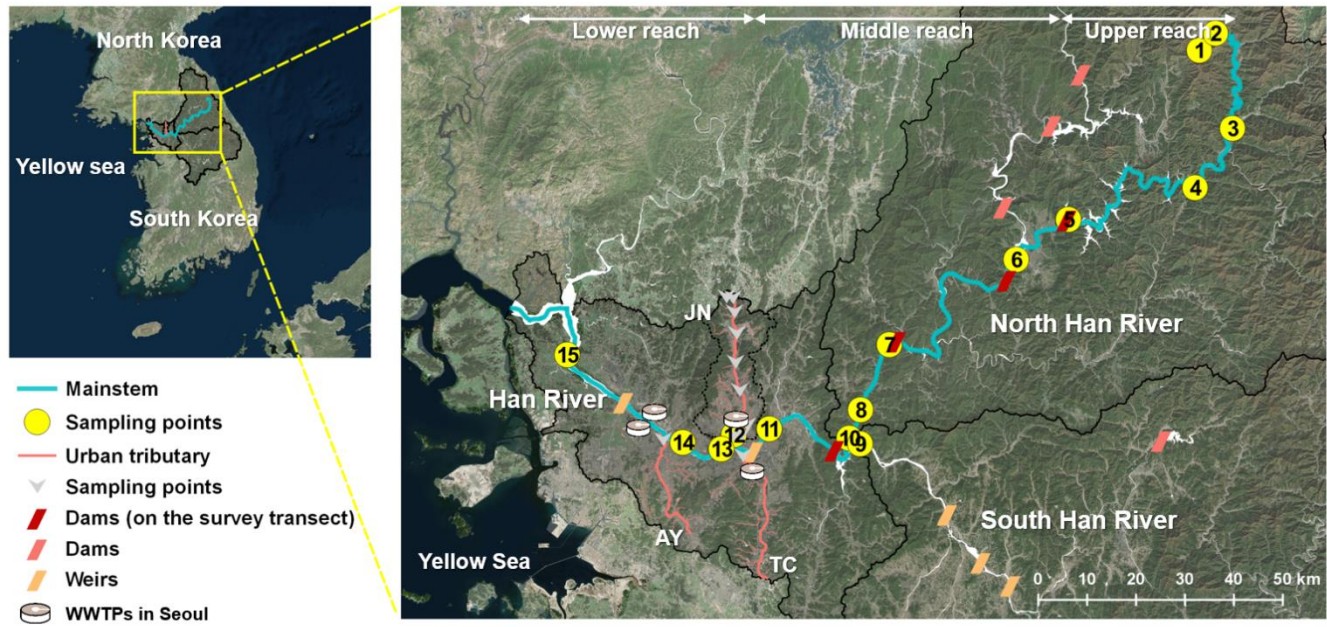

**Figure 1.** Study map showing sampling sites along the mainstem and three urban tributaries (JN, TC, and AY) in the Han River basin, South Korea.

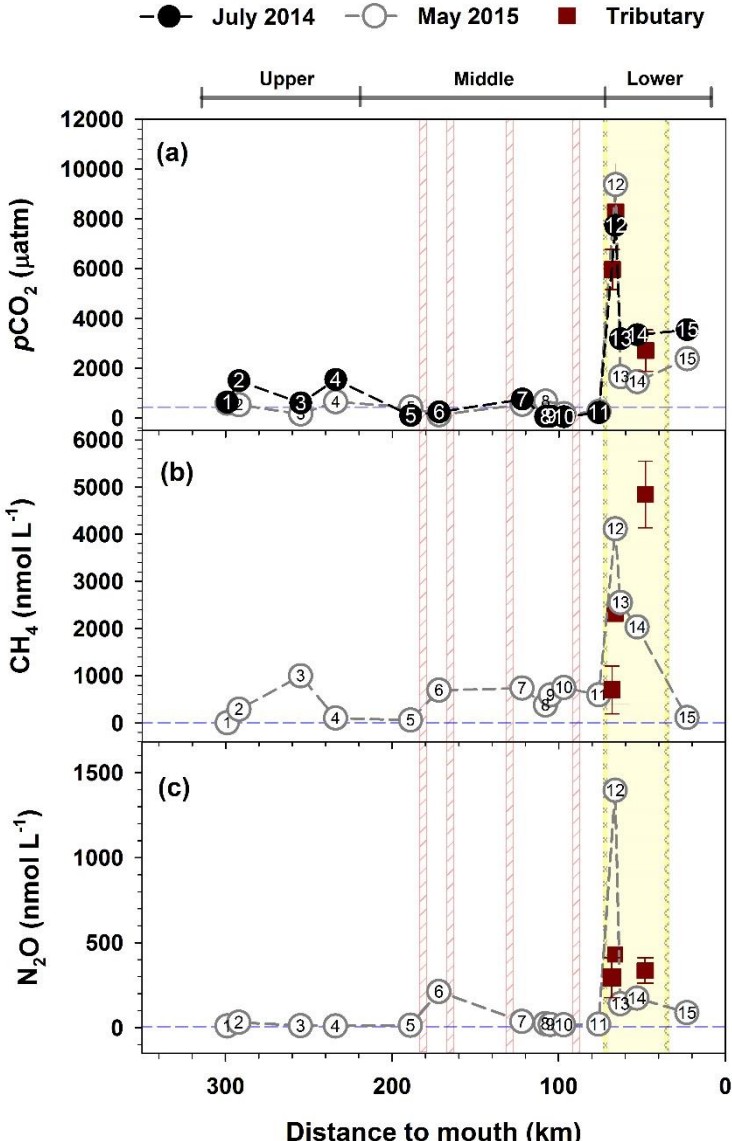

**Figure 2.** Spatial variations in $pCO_2$ (a), $CH_4$ (b), and $N_2O$ (c) measured at 15 sites and 3 urban tributaries along the Han River. $pCO_2$ was measured in two basin-wide surveys – July 2014 and May 2015 (May 2015 data are modified from Yoon et al., 2017), but $CH_4$ and $N_2O$ were measured only in May 2015. Additional measurements at the urban tributary JN (HR12; n = 13) and two other tributaries, TC (n = 4) and AN (n = 2) are indicated by brown squares with standard deviations. Dashed horizontal lines denote the mean value of atmospheric equilibrium for $pCO_2$ (435 µatm), $CH_4$ (3.3 nmol $L^{-1}$), and $N_2O$ (6.5 nmol $L^{-1}$). Four vertical lines on the middle reach indicate the location of dams. The yellow shade demarcated by two spiral lines indicate the lower-reach section separated by two submerged weirs.

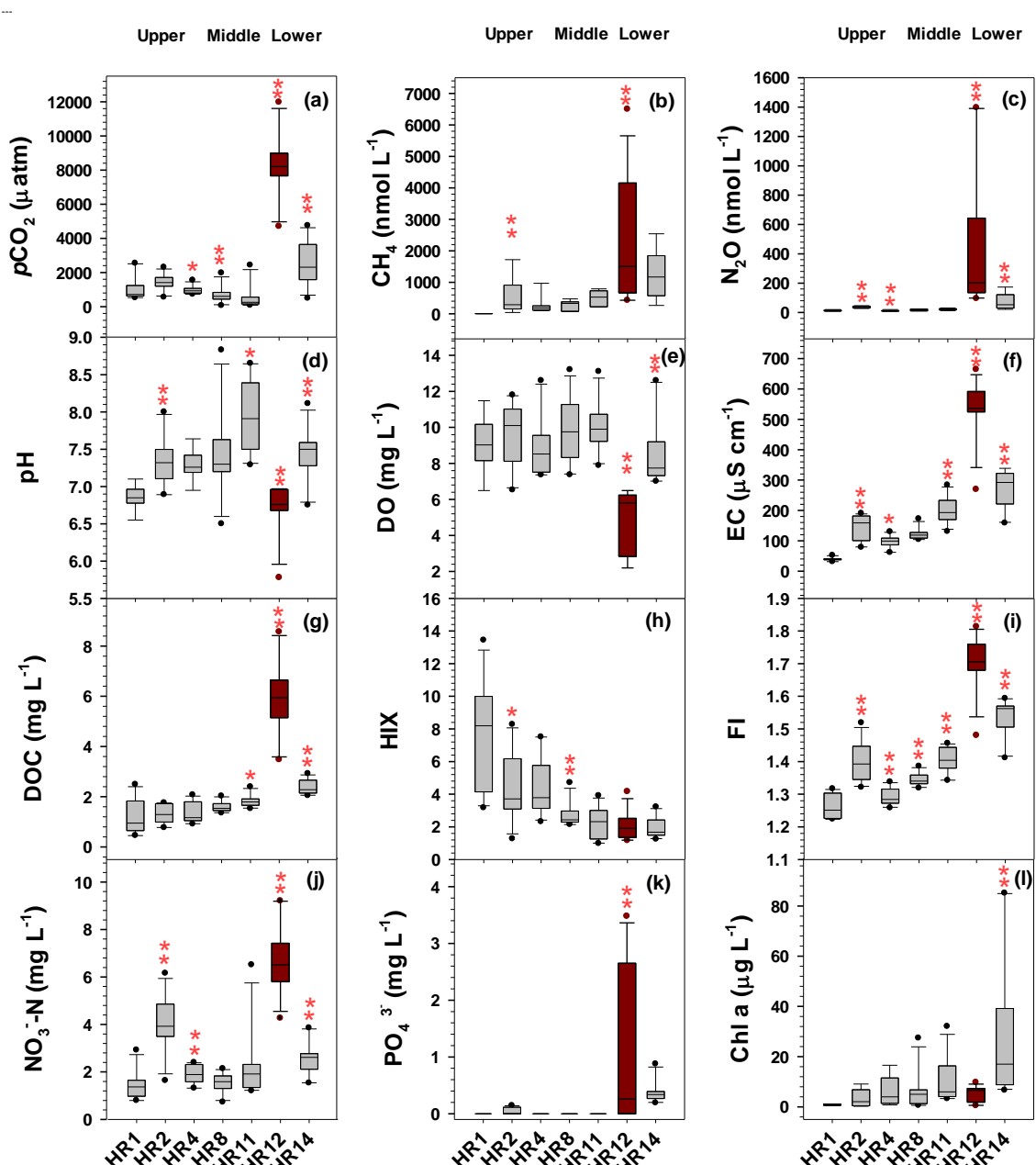

**Figure 3.** Box plots showing spatial variations in three GHGs (a−c), in situ measurements (d−f), DOC and optical characteristics (g−i), and concentrations of $NO_3^-$−N (j), $PO_4^{3-}$ (k), and Chl $a$ (l) measured at 6 monthly monitoring sites (July 2014−July 2015) and an urban tributary outlet (HR12; May 2015−December 2017). A significant downstream change between two successive sites is indicated by one or two asterisks placed on top of the downstream site (P < 0.05 or P < 0.01, respectively; Mann-Whitney test).

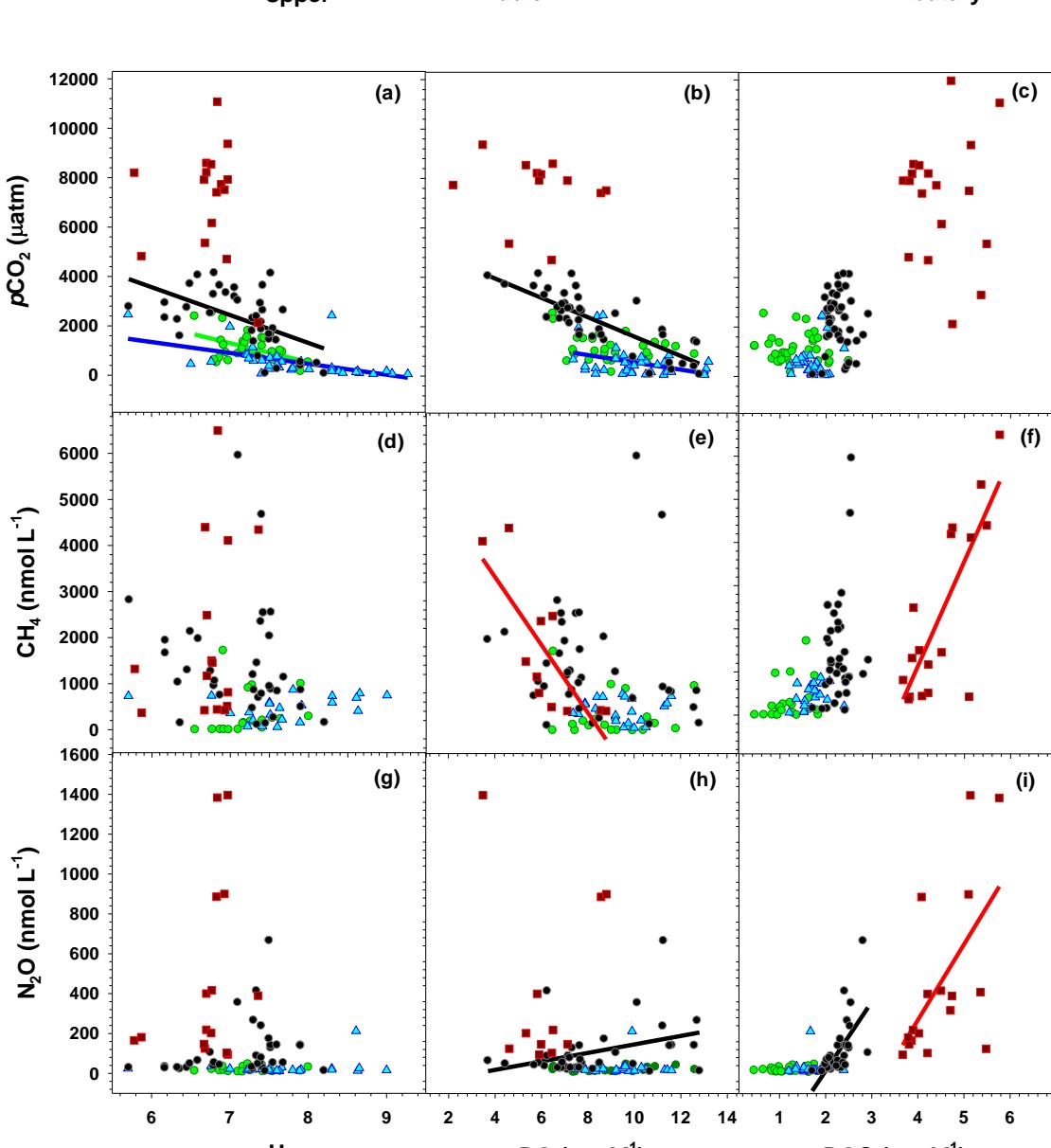

**Figure 4.** Relationships between water quality (pH, DO, and DOC) and dissolved concentrations of three GHGs ($pCO_2$, $CH_4$, and $N_2O$) measured in the Han River basin. Regression analysis was conducted with data aggregated for each of the upper, middle, and lower reaches, and the group of the three urban tributaries (TC, JN, and AY). Only significant ($P < 0.05$) relationships are indicated by the regression line through the plot.

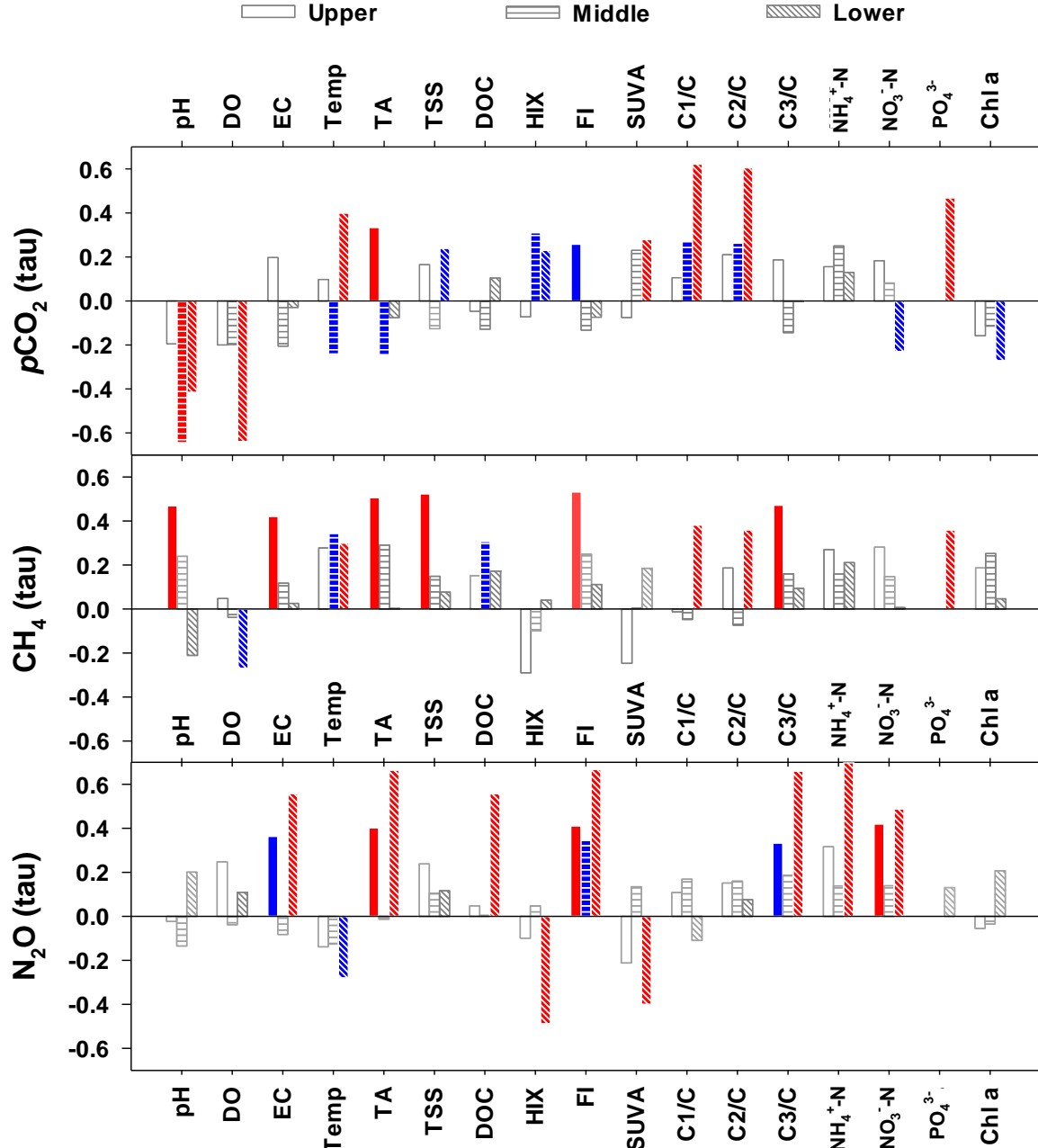

**Figure 5.** Kendall rank correlations (tau) between GHGs and water quality components measured in the upper, middle and lower reaches of the Han River. Significant correlations at P < 0.05 and P < 0.01 are indicated by blue and red, respectively. Please note that correlation analysis results were not available for $PO_4^{3-}$ in the upper and middle reaches due to many values below detection limit.

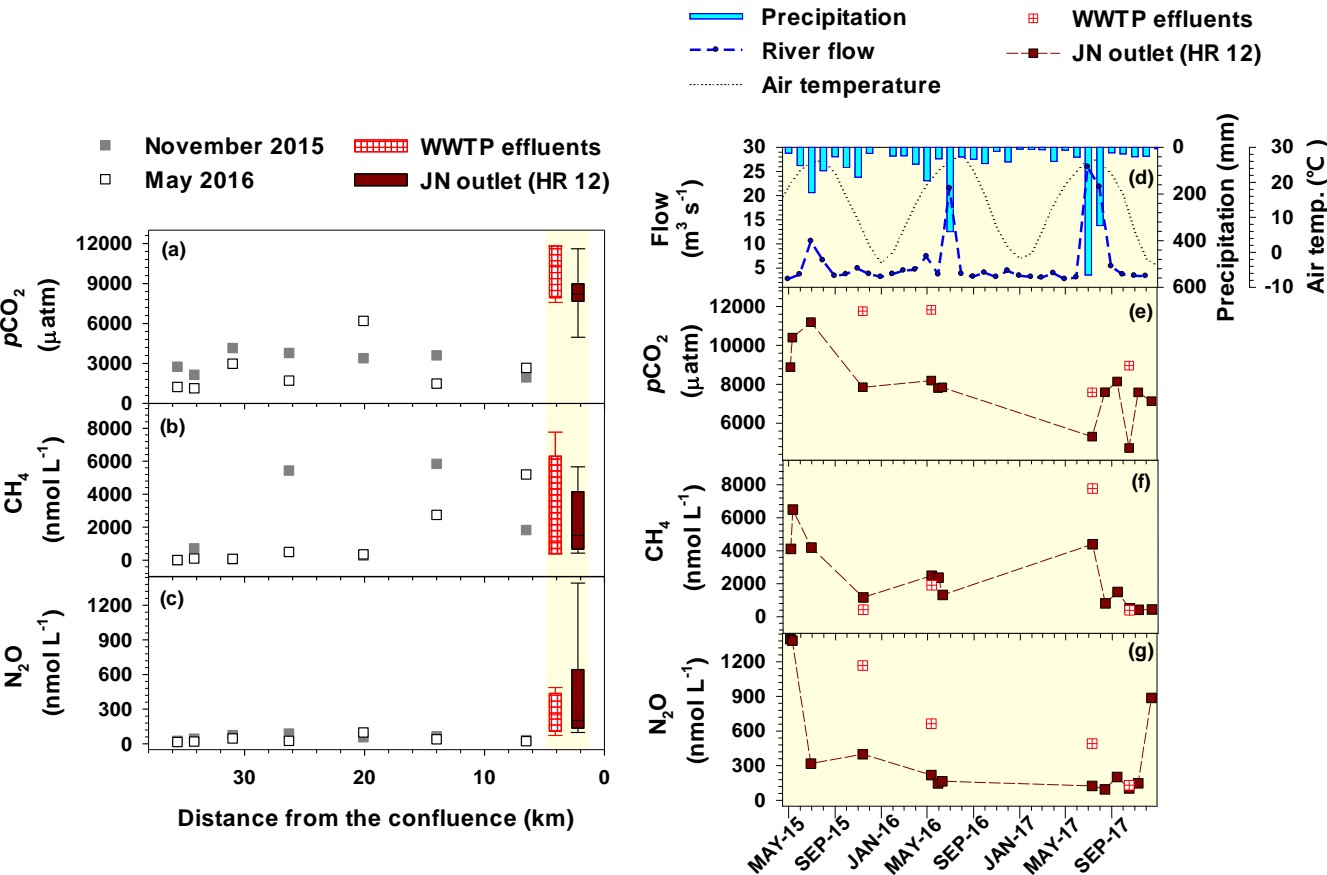

**Figure 6.** Longitudinal variations in $pCO_2$ (a; data reported in Yoon et al., 2017), $CH_4$ (b), and $N_2O$ (c) along the urban tributary (Joongnang River: JN) surveyed in November 2015 and May 2016 and temporal variations in hydroclimatic conditions (d; monthly precipitation and mean temperature and flow), $pCO_2$ (e), $CH_4$ (f), and $N_2O$ (g) monitored at the JN outlet (HR12) and WWTP effluents from May 2015 to December 2017. Box plots shown on the left panel summarize the time series data obtained from the WWTP effluents and tributary outlet, as displayed on the right panel. Weather data were obtained from an automatic weather station adjacent to HR12, while flow was measured at a bridge upstream of HR12.

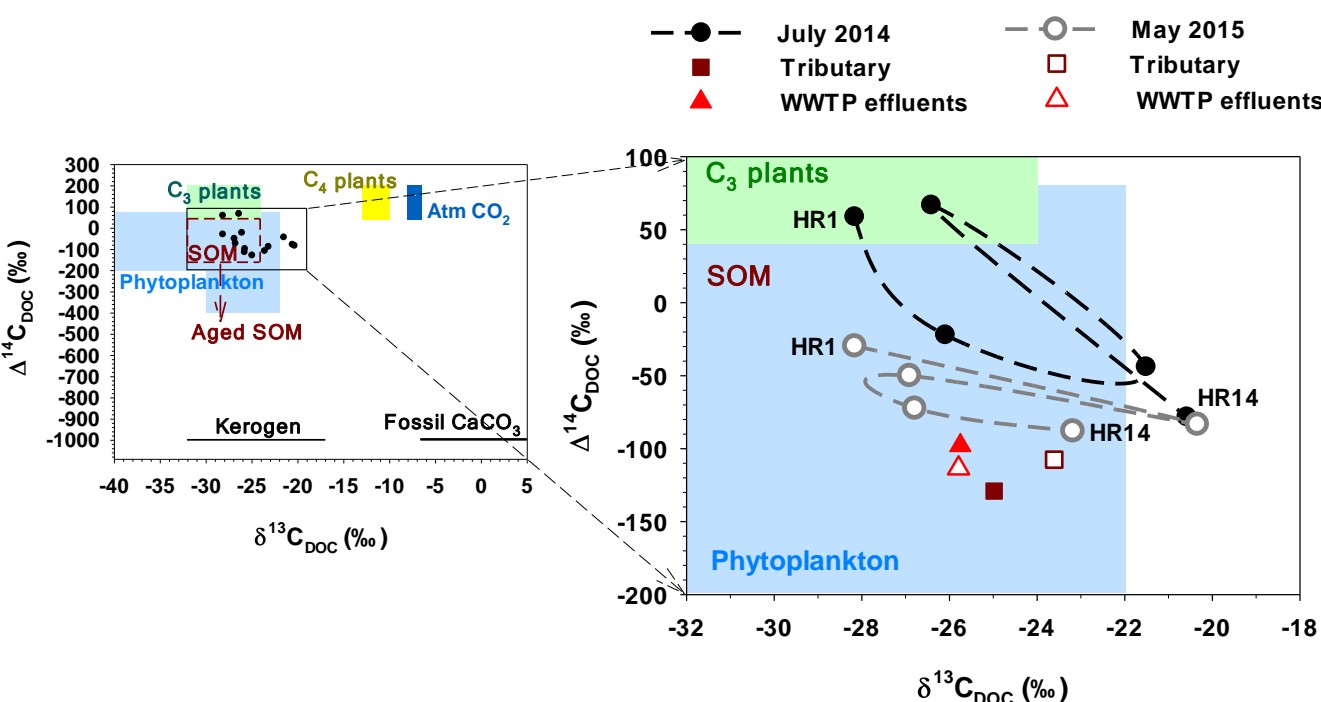

**Figure 7.** $\delta^{13}C$ and $\Delta^{14}C$ in DOM sampled at five Han River mainstem sites, an urban tributary, and a WWTP discharge. Mainstem (HR1, HR4, HR8, HR11, and HR14) and tributary (HR12) samples were collected in July 2014 and May 2015, whereas two WWTP effluent samples were obtained in June 2014 and November 2015. Reference values for DOM sources presented on the left plot are from Marwick et al. (2015) and references therein.

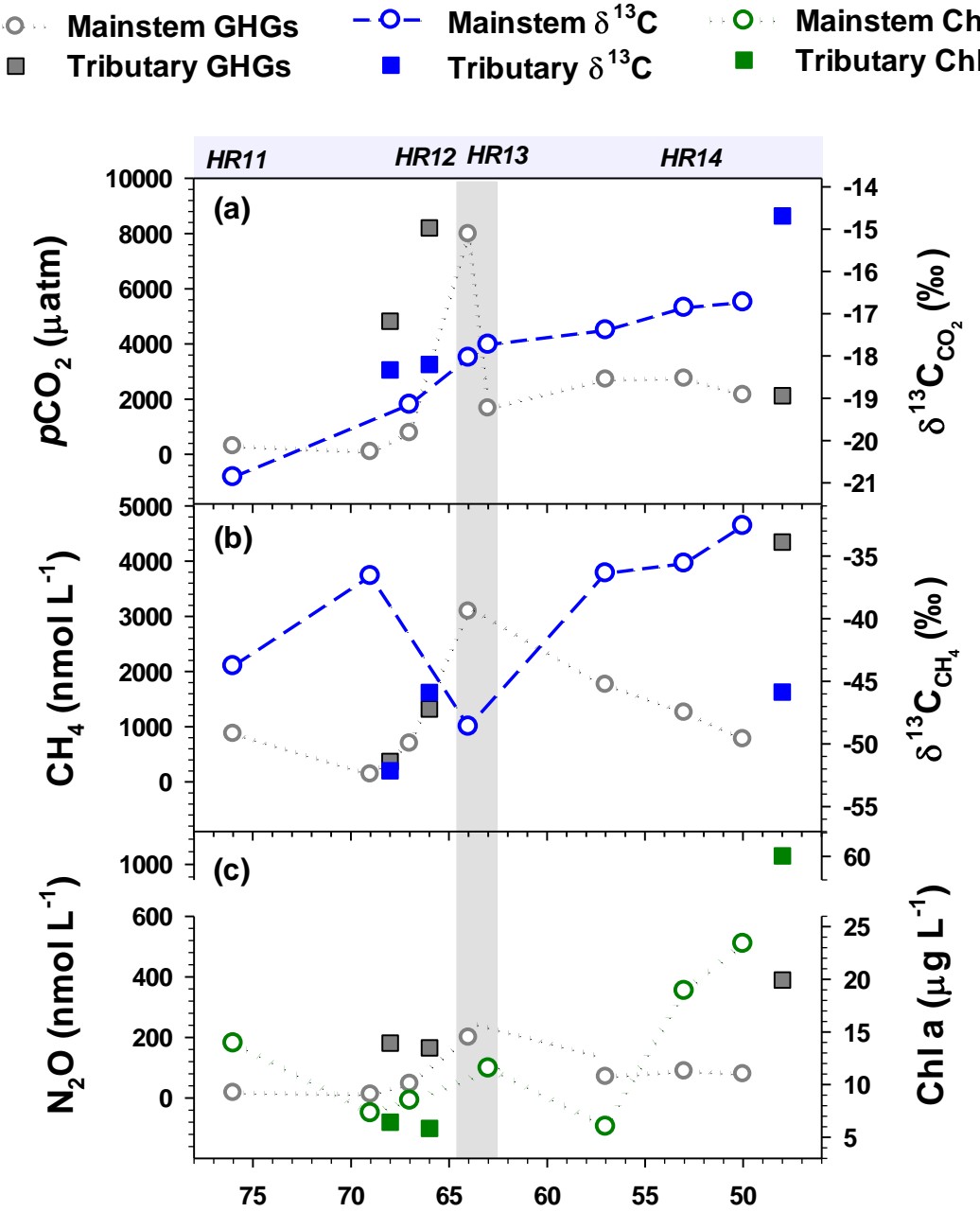

**Figure 8.** Spatial variations in $pCO_2$ and $\delta^{13}C_{CO2}$ (a), $CH_4$ concentrations and $\delta^{13}C_{CH4}$ (b), and $N_2O$ and Chl *a* concentrations (c) measured at 8 mainstem sites and 3 urban tributary outlets during a cruise expedition (June 2016) along the lower reach of the Han River. The grey shaded area indicates the mainstem section receiving the inflow from the urban tributary JN (HR12). The four vertical dashed lines denote the locations of the four monthly monitoring sites along the lower reach. Data of $pCO_2$ and $\delta^{13}C_{CO2}$ were modified from Yoon et al. (2017)