# Peer review of "Longitudinal discontinuities in riverine greenhouse gas dynamics generated by dams and urban wastewater"

_Biogeosciences, 2018_

## Referee Comment (RC1) · Anonymous Referee #1 · 11 Jul 2018

The manuscript bg-2018-278 by Jin et collaborators explores Vannote's (1980) river continuum concept in the light of river damming and urban effluents. The dataset is consistent and the statistical approaches (nonparametric tests) seem appropriate. Nevertheless, I would recommend replacing the fitting ($R^2$, p-value) by discriminant/cluster analyses. There is no physical meaning in $R^2$ values that, despite the $p < 0.001$, evidence weak correlations (coefficients of determination $\sim < 50\%$). Those plots are more suitable for discriminating spatial variability than fitting meaningless polynomials. On the other hand, the authors should also consider references for broadening the systemic understanding of the focused problem. I recommend to the authors to: 1) Explore/discuss your data under the Riverine Ecosystem Synthesis (Thorp, J.H., J.E. Flotemersch, M.D. De-long, A.F. Casper, M.C. Thoms, F. Ballantyne, B.S. Williams, B.J. O'Neill, C.S. Haase. 2010. Linking Ecosystem Services, Rehabilitation, and River Hydroge-omorphology. BioScience 59(1): 67-74. https://doi.org/10.1525/bio.2010.60.1.11), which extends the river continuum approach with the flood pulse and space-time scaling; 2) Explore/discuss your data under the ecohydrology perspective (Bergier, I., Ramos, F.M. & Bambace, L.A.W. Environ Monit Assess (2014) 186: 5985. https://doi.org/10.1007/s10661-014-3834-2) that regards the land-use in the landscape as fueling GHG emissions; and 3) Finally, also consider the study provided in Abe et al (2009) (https://www.tandfonline.com/doi/abs/10.1080/03680770.2009.11902248) regarding wastewater, algal bloom and GHG emissions from dams.

---

## Referee Comment (RC2) · Anonymous Referee #2 · 13 Jul 2018

General comments.

The manuscript bg-2018-78: "Longitudinal discontinuities in riverine greenhouse gas dynamics generated by dams and urban wastewater" by Hyojin Jin et al provides an interesting study about the basin-scale patterns of the three major greenhouse gases ($CO_2$, $CH_4$, $N_2O$) in a highly urbanized watershed. The study outlines the importance of dams and wastewater treatment plant with regards to the river continuum concept (Vannote et al., 1980) and could be significant in the field of biogeochemistry of highly human-managed watersheds. The study show that dams creates discontinuities in the hydrological continuum, which favored aquatic autotrophy and then the release of $CH_4$ and $N_2O$ from the sediments. Wastewater treatment plants release high concentration of the three GHGs and replenished labile riverine pool of DOM, fueling the river heterotrophy. The dataset is very large in both spatial and temporal scales, methods and sampling design are appropriate, figures are of high quality and the study is well documented. Statistical analysis are also appropriate but are only bivariate analysis and thus I think that it would be interesting to explore the dataset further by doing multivariate analysis (see my comments below). Overall, I support publication of this manuscript and below are some more detailed comments.

Specific comments.

L.40-43. Definitely, there is a lack of direct $pCO_2$ measurements in Asia and Africa, but this is also true in Europe and America since the GLORICH database used in global $CO_2$ synthesis originates from pH/TA/temperature calculations (Hartmann et al., 2014). $pCO_2$ calculated from pH/TA/temperature is strongly overestimated notably in low, buffered and high DOC waters such as boreal and tropical rivers, which strongly contribute to the global $CO_2$ degassing (Abril et al., 2015). In addition, taking into account that wetlands and flooded land are now recognized as significant to the regional and global carbon budget (Abril et al., 2014; Abril and Borges, 2018), we are still far to obtain a precise carbon budget at the global scale. Therefore, if authors want to introduce global $CO_2$ synthesis, I would suggest specifying the above information.

L.43-45. I would suggest to add this reference where $CO_2$, $CH_4$ and $N_2O$ have been measured simultaneously in the Zambezi River (Teodoru et al., 2015).

L.90-97. In my opinion, those sentences belong to the study site section.

L.115-118. I would suggest to show land use on the map of the figure 1 (see my comments below for the figure 1).

L.135-136. According to Fig. 1, JN transect is an highly urbanized tributary but authors wrote that in this transect there is a forested headwater. This seems paradoxical to me (cf my comments of the Fig. 1).

L.189. Please refer to Gran (1952).

L.189. Usually, electro-titration of TA with the Gran method used 0.1N HCl as titrant.

L.204. Please insert period after "parameter".

L.218-220. There are two forested streams (one on the JN transect and one on the main transect, right?). To avoiding any confusion, I would suggest to specify between brackets the station name. Otherwise, the reader always needs to search this information in other figures or tables. I would suggest doing the same for the remainder of the text.

L.225-227. Visualizing the Fig.S1, I am not totally agree with author's comment. At the HR14 sampling station, $N_2O$ and $CH_4$ seemed affected by season (notably spring and summer), as well at the HR2 and HR4 sampling stations where $CH_4$ seemed affected by summer/winter seasons. In order to determine if seasons significantly affects GHG concentrations at a given station, I would suggest performing a Kruskall-Wallis test accompanied with a Dunn's test in order to accounting for the multiple comparison.

L.228-237. I think that it would be interesting to know if decrease/increase described in this paragraph with the Figure 3 are statistically significant. For that, I would recommend performing a Mann-Whitney test between stations that are following each other's (testing HR1-HR2, then HR2-HR4…etc). In addition, I would suggest adding Mann-Whitney test results in the Figure 3.

L.241-252. To understand basin-scale controls on $CO_2$, $CH_4$, $N_2O$ concentrations, authors explore their dataset by doing bivariate analysis (e.g., Kendall rank correlation) between either $CO_2$, $CH_4$ or $N_2O$ and each water quality parameter for the lower/middle/upper reach. This statistical test is appropriate but I think that a multivariate analysis (as PCA, may be associated with a cluster analysis of variable) with all parameter for each lower/middle/upper reach would be also very interesting. Another possible PCA would be a PCA biplot (graph of individuals and variables together), with all the dataset, in order to see where the lower/middle/upper reach points are situated with regards to the variability of the dataset. I supposed that a multivariate analysis will learn the authors more about the variability of the dataset, and how control patterns of $CO_2$, $CH_4$ or $N_2O$ evolved from upstream (upper reach) to downstream (lower reach). In addition, it will give information about which variables are important to describe the variability of the dataset. What do you think?

L.276-278. Please add per mil symbols.

L.301. Richey et al (1988) is somewhat outdated, please add Abril et al. (2014).

L.304. "…regulated river system". May the authors add references?

L.304-308. This is a 6 lines sentence, quite difficult to follow, please consider revising the sentence.

L.336. Please add references about methanotrophy in water column of lake (e.g., Morana et al., 2015; Roland et al., 2017).

L.343-346. In dam water column, you mentioned previously that the enrichment in $CH_4$ originates from anaerobic conditions in organic-rich sediments. Usually, in strictly anaerobic conditions as occur for the methanogenesis, denitrification in the sediment is 'complete' producing $N_2$ gas and not $N_2O$. However, water column is oversaturated in $N_2O$. How do you explain this? Did you measure GHG, $O_2$ or $NH_4^+/NO_3^-$ in the profile of the water column?

L.355-359. This is a 5 lines sentence, quite difficult to follow, please consider revising the sentence. In addition, it is not clear to me, all the data presented in this sentence originates from Yoon et al (2016)? Please, specify.

L.369.370. Authors mentioned that the amount of $CH_4$ and $N_2O$ discharged from the WWTP appeared to drive the magnitude and temporal variability of the tributary inputs to the lower reach. When I observed the figure 6, this is necessary true for $N_2O$, but not necessary true for $CH_4$. Indeed, $CH_4$ increased way before the appearance of the WWTP, and the two points of Nov 2015 and May 2016 that are very different suggest a high temporal variability that could explain $CH_4$ concentrations measured at HR12. Do not you think that there is another source of $CH_4$ than WWTP for this tributary?

L.385.402. In this paragraph, could you explain the spatial longitudinal pattern of $\delta^{13}C$-DOC?

L.388. Did you mean 72 among 695 or did you mean 72%? Please, specify.

L.403. What does RKM term means? Please, specify?

L.403-416. All the statements you mentioned in this paragraph are maybe true but remain unclear to me. To improve this paragraph, I think that you need to better identify inputs and processes playing a role in the variability of $\delta^{13}C$-$CO_2$ signature in the studied river.

First, I am partially agree with the first sentence because dissolution of carbonates is $CaCO_3+CO_2+H_2O \rightarrow 2HCO_3^- + Ca^{2+}$. Thus, dissolution of carbonates will not influence $\delta^{13}C$-$CO_2$ signature but will influence $\delta^{13}C$-$HCO_3^-$ and thus $\delta^{13}C$-DIC signature (e.g., Deirmendjian and Abril, 2018).

Second, you mentioned $\delta^{13}C$-$CO_2$ originating from riverine organic matter degradation. So, did you mean riverine organic matter coming from aquatic autotrophy? or riverine organic matter coming from soil and groundwaters leaching that is degraded in river? Because both sources have a distinct $\delta^{13}C$-DOC signature. Thereafter you compared $\delta^{13}C$-$CO_2$ value originating from riverine organic matter degradation with $\delta^{13}C$-$CO_2$ value originating from lakes to highlight the fact that there are other processes than bacterial degradation to explain the variability $\delta^{13}C$-$CO_2$ in your dataset. According to the $\delta^{13}C$-$CO_2$ value of the lake, in this lake a high proportion of the $CO_2$ originates from terrestrial degradation of DOC from C3 plants. So when you compared $\delta^{13}C$-$CO_2$ originating from degradation of C3 plants with $\delta^{13}C$-$CO_2$ originating from degradation of riverine phytoplankton, the difference you observed originates from a different $\delta^{13}C$ signature of DOC. In addition, the lake you mentioned is boreal, so, you cannot compared boreal system with yours. Here, I supposed that you need to focus on your data to explain that there is another process than bacterial degradation involved in the shifting of $\delta^{13}C$-$CO_2$ in your dataset. At the same transect, figure 7 shows $\delta^{13}C$-DOC values being very lower than the corresponding $\delta^{13}C$-$CO_2$ values. As during degradation of DOC, the main source of $CO_2$ in large river (Hotchkiss et al., 2015), fractionation do not occur (Amundson et al., 1998), the difference between $\delta^{13}C$-$CO_2$ and $\delta^{13}C$-DOC is due to another process than bacterial degradation of DOC.

The next sentence, you refer to study dealing with $\delta^{13}C$-DIC, but you did not measure this parameter. This brings misunderstanding, because you cannot compare the evolution of the $\delta^{13}C$-$CO_2$ with $\delta^{13}C$-DIC. Indeed, I mentioned previously that dissolution of carbonates would increase $\delta^{13}C$-DIC but not $\delta^{13}C$-$CO_2$. In addition, we do not know where comes the carbonates could come from. Is there some carbonate precipitation in some part of the studied river, and then dissolution in other part? Did carbonates come from weathering of carbonates in land?

Last sentence remain also unclear to me. I am agree with the first statement that photosynthesis will increase $\delta^{13}C$ signal of $CO_2$ by a preferential using of lighter $^{12}CO_2$. In the second statement, you mentioned that dissolution of atmospheric $CO_2$ with a heavier signature (approximately -8‰) will increase $\delta^{13}C$ signal of $CO_2$ in the water. This is true when riverine water is undersaturated in $CO_2$ with respect to the atmosphere equilibrium, which enables atmospheric $CO_2$ to enter the water. However, in oversaturated water, due to water-air $CO_2$ gradient, no $CO_2$ (light or heavy) enter the water but on the contrary, $^{12}CO_2$ and $^{13}CO_2$ are both degassed to the atmosphere. In addition, $^{12}CO_2$ degassed to the atmosphere at a faster rate than $^{13}CO_2$ because $p^{12}CO_2$ gradient between air and water is larger than the $p^{13}CO_2$ gradient, this process will increase $\delta^{13}C$-$CO_2$ and $\delta^{13}C$-DIC signature in the riverine water (e.g., Deirmendjian and Abril, 2018; Polsenaere and Abril, 2012). What is the influence of degassing in the variability of $\delta^{13}C$-$CO_2$ in

your transect? In the third statement, you mentioned the preferential used of $^{12}CO_2$ by heterotopic bacteria, but how heterotrophic bacteria can used $CO_2$? Please, clarify.

L.417-418. I supposed that you refer at the isotopic fractionation due to the thermodynamic equilibrium between $CO_2$ and $HCO_3^-$? However, you cannot status only with this information that in your studied river $\delta^{13}C$-DIC signature will be 10‰ higher than $\delta^{13}C$-$CO_2$ signature. Indeed, Equation of $\delta^{13}C$-DIC is

$$\delta^{13}C\text{-DIC} = (\delta^{13}C\text{-}CO_2^* \times [CO_2^*] + [HCO_3^-] \times \delta^{13}C\text{-}HCO_3^- + [CO_3^{2-}] \times \delta^{13}C\text{-}CO_3^{2-}) / ([CO_2^*] + TA)$$

The signature of $\delta^{13}C$-DIC depends thus on complex interplays between initials concentration of each dissolved inorganic parameter as well as their signature, then processes producing or consuming DIC (primarily photosynthesis, degassing, respiration, weathering), and the isotopic thermodynamic equilibrium between each compounds.

L.418.420. However, in the first part of the figure $\delta^{13}C$-$CO_2$ increased at the same rate as in the second part of the figure but without any increase in Chl a. Please, explain.

L.421.422. Can you explain the difference in $\delta^{13}C$-$CO_2$ between tributaries and main stem?

L.431.432. Does degassing of $CH_4$ to the atmosphere could have an impact on the upstream-downstream decrease of $CH_4$?

L.445. To conclude, do you have any recommendations for politician, river managers and stakeholders to improve water quality and reducing GHG concentrations in highly urbanized watershed?

Tab. 1. I would suggest adding a left column to specify upper/middle/lower reach.

Fig. 1: I am not aware if a land use database exists for South Korea, but if such a database exists, I would recommend adding the land use in the map of the Figure 1, particularly to visualize where croplands, forest and cities are located. In addition, to visualize the proportion of croplands, forest and cities in the studied catchment. I would also suggest adding the forested headwater from the JN transect in another color than the other points of the JN transect. Indeed, JN transect is considered by the authors as an urban transect, and thus, this is strange to associate an urbanized river with a forested headwater. Perhaps, authors could also apply a different typology for the sampling points, with for example, one color for forested streams, one for agricultural streams…It would be easier to visualize sampling points in the map of the Figure 1. Please, also add metric scale on the map.

Fig. 2. I would suggest specifying upper/middle/lower reach in the figure, perhaps at the top of the figure.

Fig. 3. I would suggest specifying upper/middle/lower reach in the figure, perhaps at the top of the figure.

Fig.4. I would suggest to specify which tributaries belong to the red points (JN? TC? AN? or this is just HR12?)

Fig.8. I would suggest to specify sampling stations names on the graphs.

---

## Referee Comment (RC3) · Anonymous Referee #2 · 13 Jul 2018

I forgot to put the references that I used in my comments.

References are the following

Abril, G., Borges, A.V., 2018. Carbon leaks from flooded land: do we need to re-plumb the inland water active pipe? Biogeosciences Discuss. 2018, 1–46. https://doi.org/10.5194/bg-2018-239 Abril, G., Bouillon, S., Darchambeau, F., Teodoru, C.R., Marwick, T.R., Tamooh, F., Ochieng Omengo, F., Geeraert, N., Deirmendjian, L., Polsenaere, P., Borges, A.V., 2015. Technical Note: Large overestimation of pCO2 calculated from pH and alkalinity in acidic, organic-rich freshwaters. Biogeosciences 12, 67–78. https://doi.org/10.5194/bg-12-67-2015 Abril, G., Martinez, J.-M., Artigas, L.F.,

[Figure]

Moreira-Turcq, P., Benedetti, M.F., Vidal, L., Meziane, T., Kim, J.-H., Bernardes, M.C., Savoye, N., Deborde, J., Souza, E.L., Albéric, P., Landim de Souza, M.F., Roland, F., 2014. Amazon River carbon dioxide outgassing fuelled by wetlands. Nature 505, 395–398. https://doi.org/10.1038/nature12797 Amundson, R., Stern, L., Baisden, T., Wang, Y., 1998. The isotopic composition of soil and soil-respired CO 2. Geoderma 82, 83–114. Deirmendjian, L., Abril, G., 2018. Carbon dioxide degassing at the groundwater-stream-atmosphere interface: isotopic equilibration and hydrological mass balance in a sandy watershed. J. Hydrol. Gran, G., 1952. Determination of the equivalence point in potentiometric titrations of seawater with hydrochloric acid. Ocean. Acta 5, 209–218. Hartmann, J., Lauerwald, R., Moosdorf, N., 2014. A brief overview of the GLObal RIver CHemistry Database, GLORICH. Procedia Earth Planet. Sci. 10, 23–27. Hotchkiss, E.R., Hall Jr, R.O., Sponseller, R.A., Butman, D., Klaminder, J., Laudon, H., Rosvall, M., Karlsson, J., 2015. Sources of and processes controlling CO2 emissions change with the size of streams and rivers. Nat. Geosci. 8, 696–699. Morana, C., Borges, A.V., Roland, F.A.E., Darchambeau, F., Descy, J.-P., Bouillon, S., 2015. Methanotrophy within the water column of a large meromictic tropical lake (Lake Kivu, East Africa). Biogeosciences 12, 2077–2088. Polsenaere, P., Abril, G., 2012. Modelling CO2 degassing from small acidic rivers using water pCO2, DIC and $\delta$13C-DIC data. Geochim. Cosmochim. Acta 91, 220–239. https://doi.org/10.1016/j.gca.2012.05.030 Roland, F.A., Darchambeau, F., Morana, C., Bouillon, S., Borges, A.V., 2017. Emission and oxidation of methane in a meromictic, eutrophic and temperate lake (Dendre, Belgium). Chemosphere 168, 756–764. Teodoru, C.R., Nyoni, F.C., Borges, A.V., Darchambeau, F., Nyambe, I., Bouillon, S., 2015. Dynamics of greenhouse gases (CO 2, CH 4, N 2 O) along the Zambezi River and major tributaries, and their importance in the riverine carbon budget. Biogeosciences 12, 2431–2453. Vannote, R.L., Minshall, G.W., Cummins, K.W., Sedell, J.R., Cushing, C.E., 1980. The river continuum concept. Can. J. Fish. Aquat. Sci. 37, 130–137.

---

## Referee Comment (RC4) · Anonymous Referee #3 · 23 Jul 2018

Jin and co-authors present an extensive dataset of greenhouse gas (GHG) measurements along a human-impacted river in Korea. The river is divided in three sections: the upper reach which is characterised by forest and agricultural land use, the middle reach which is impacted by multi-purpose dams and the lower reach which is influenced by wastewater discharge of the city of Seoul. Significant discontinuities in the GHG concentrations were found in the dam and sewage impacted reaches. Although the conclusions are not very surprising, the importance of this manuscript is the comprehensive dataset created by the authors, which provides a lot of quantitative information for larger-scale overview articles.

General comments

In the introduction, you often mention that previous studies looked at only a single anthropogenic factor. It took me a second reading before I distinguished the two anthropogenic factors, dams and sewage, as spatially distinct along the river (middle and lower reach). Even though it might be a slight over-simplification, it might help the reader if you make it more explicit (similar to the second sentence of previous paragraph). Your many sites and tributaries can become confusing, but framing it as 'natural', 'dams' and 'urban/sewage' would help to keep track.

You have a tendency to make complicated sentences because you want to include all your reasoning or justifications in one sentence. While these sentences were grammatically correct, they are really hard to read. Be critical to sentences which are more than 4 lines and consider splitting them up. I will indicate a few of those sentences in the detailed comments.

Specific comments

L. 16: I have difficulties with calling the dams and sewage primary controls, because I perceive the term 'primary' as the 'first', while the human impact is actually superimposed on the natural dynamics. I would suggests changing it to "major controls". Also, the effects are not the controlling the GHG dynamics. "... to investigate the influence of dams and urban water pollution on GHG dynamics ..."

L. 28: might (without e at the end)

L. 112: Add the length of the river

L. 115-118: Split over two sentences. One about major land use, one about the metropolitan area.

L.127: What is the treatment level of the three WWTPs.

L. 138-140: What are the observation dates/month & year?

L. 219: It was not clear to me where the agricultural stream and forested headwater stream belong to. Are they both part of the upper reach? Also the submerged weirs is not clear to which section they belong. It felt like you are jumping up and down along the river in the description of the longitudinal variations. Try to be consistent in describing each parameter from upper reach over middle reach to lower reach.

L. 225: replace "less impacted upstream or downstream reaches" with "compared to the upper and lower reaches". All of the reaches are impacted, just in a different way.

L. 225-227: This is a complicated sentence. Consider splitting it up (especially the explanation for sites HR8 and HR11).

L. 233: What is the water discharge ratio between the tributary and the main river?

L. 238: This is a complicated sentence. "When we pooled the measurements for the whole river basin, at least two of the GHG's exhibited significant ..."

L. 260: How can the WTTP effluents and tributary reach values of the upstream river. Consider rephrasing.

L. 261: the large scatter (without s)

L. 273: "though" doesn't seem the correct word.

L. 304-309: Very long and complicated sentence with lots of subsentences.

L. 408: Could the composition of the respired organic material be responsible for the variation in $\delta$13C? I expect very little C4 plants in Canada, which is consistent with the very low $\delta$13C values. If you have more variation in C3-C4 plants throughout your catchment, then you would expect to see that change reflected in the riverine C.

Figure 2: Could you indicate the three different reaches in the graphs?

---

## Author Comment (AC1) · 19 Aug 2018

**Author response to RC1**

The manuscript bg-2018-278 by Jin et collaborators explores Vannote's (1980) river continuum concept in the light of river damming and urban effluents. The dataset is consistent and the statistical approaches (nonparametric tests) seem appropriate. Nevertheless, I would recommend replacing the fitting (R2, p-value) by discriminant/cluster analyses. There is no physical meaning in R2 values that, despite the p<0.001, evidence weak correlations (coefficients of determination ～<50%). Those plots are more suitable for discriminating spatial variability than fitting meaningless polynomials.

**<Response> We thank you for your positive evaluation of our manuscript. Results of reach-based data clustering (Fig. 4) and PCA (Fig. S3) have been included in the revised manuscript in response to your and the second reviewer's suggestions. Descriptions of used statistical analyses and data interpretations have been provided in relevant sections (Lines 257-264:** When all measurements of three GHGs and water quality were pooled for the whole river basin, at least one of three GHGs exhibited an overall negative relationship with pH ($pCO_2$) and DO ($pCO_2$ and $CH_4$) and a positive relationship with DOC (all three GHGs) (Fig. 4). Regression analysis conducted with separate data sets clustered for each of three reaches and urban tributaries showed several significant negative or positive relationships (Fig. 4). A positive relationship between DO and $N_2O$ in the lower reach was noticeable compared with no significant relationship found for the other reaches. Reach-specific clustering of data was also found on a PCA scatter plot with two primary components accounting for 57.5% of variations (Fig. S3). While the upper and middle reach data were overlapped considerably on the PCA scatter plot (the upper reach with a wider scatter), the majority of the lower reach data were separated from the overlap of the upper and middle reaches.**).**

[Figure]

**Figure 4.** Relationships between water quality (pH, DO, and DOC) and dissolved concentrations of three GHGs ($pCO_2$, $CH_4$, and $N_2O$) measured in the Han River basin. Regression analysis was conducted with data clustered for each of the upper, middle, and lower reaches, and three urban tributaries (TC, JN, and AY). Only significant (P < 0.05) relationships are indicated by the regression line through the plot.

[Figure]

Fig. S3. Reach-based grouping of all measurements in the upper, middle, and lower reaches of the Han River alonng two components identified by principal component analysis (PCA).

On the other hand, the authors should also consider references for broadening the systemic understanding of the focused problem. I recommend to the authors to: 1) Explore/discuss your data under the Riverine Ecosystem Synthesis (Thorp, J.H., J.E. Flotemersch, M.D. Delong, A.F. Casper, M.C. Thoms, F. Ballantyne, B.S. Williams, B.J. O'Neill, C.S. Haase. 2010. Linking Ecosystem Services,   Rehabilitation, and River Hydrogeomorphology. BioScience 59(1): 67-74. https://doi.org/10.1525/bio.2010.60.1.11), which extends the river continuum approach with the flood pulse and space-time scaling; 2) Explore/discuss your data under the ecohydrology perspective (Bergier, I., Ramos, F.M. & Bambace, L.A.W. Environ Monit Assess (2014) 186: 5985. https://doi.org/10.1007/s10661-014-3834-2) that regards the land-use in the landscape as fueling GHG emissions; and  3)  Finally,  also  consider  the  study  provided  in  Abe  et  al  (2009) (https://www.tandfonline.com/doi/abs/10.1080/03680770.2009.11902248) regarding wastewater, algal bloom and GHG emissions from dams.

**<Response> Thanks for recommending these useful references. Two papers have been cited in L 500-503 (following sentences stressing the limitation of the conventional river continuum concept):** The observed reach-specific patterns of altered water quality and GHG dynamics provide empirical evidence for ecosystem structural and functional responses to anthropogenic changes in hydrogeomorphic patches of the fluvial landscape, which have been emphasized in recent conceptual models integrating fluvial geomorphology and ecosystem processes at the valley to reach scales (Thorp et al., 2010).

**Also in L 366-367 (following a discussion of DOC-CH4 transformation):** As noted by Bergier et al. (2014), organic wastes released from local sources might have contributed to the transformation of DOC to CH4.

---

## Author Comment (AC2) · 19 Aug 2018

**Author response to RC2**

General comments.

The manuscript bg-2018-78: "Longitudinal discontinuities in riverine greenhouse gas dynamics generated by dams and urban wastewater" by Hyojin Jin et al provides an interesting study about the basin-scale patterns of the three major greenhouse gases (CO2, CH4, N2O) in a highly urbanized watershed. The study outlines the importance of dams and wastewater treatment plant with regards to the river continuum concept (Vannote et al., 1980) and could be significant in the field of biogeochemistry of highly humanmanaged watersheds. The study show that dams creates discontinuities in the hydrological continuum, which favored aquatic autotrophy and then the release of CH4 and N2O from the sediments. Wastewater treatment plants release high concentration of the three GHGs and replenished labile riverine pool of DOM, fueling the river heterotrophy. The dataset is very large in both spatial and temporal scales, methods and sampling design are appropriate, figures are of high quality and the study is well documented. Statistical analysis are also appropriate but are only bivariate analysis and thus I think that it would be interesting to explore the dataset further by doing multivariate analysis (see my comments below). Overall, I support publication of this manuscript and below are some more detailed comments.

**<Response> We thank you for your positive evaluation of our manuscript. According to your suggestion, a multivariate analysis was carried out. Please refer to our detailed responses to your specific comments below.**

Specific comments.

L.40-43. Definitely, there is a lack of direct pCO2 measurements in Asia and Africa, but this is also true in Europe and America since the GLORICH database used in global CO2 synthesis originates from pH/TA/temperature calculations (Hartmann et al., 2014). pCO2 calculated from pH/TA/temperature is strongly overestimated notably in low, buffered and high DOC waters such as boreal and tropical rivers, which strongly contribute to the global CO2 degassing (Abril et al., 2015). In addition, taking into account that wetlands and flooded land are now recognized as significant to the regional and global carbon budget (Abril et al., 2014; Abril and Borges, 2018), we are still far to obtain a precise carbon budget at the global scale. Therefore, if authors want to introduce global CO2 synthesis, I would suggest specifying the above information.

**<Response> Some issues on potential overestimation of calculated pCO2 values, together with a growing recognition of the contribution of wetlands, have been cited as follows:**

**Lines 39-40 (recent recognition of wetlands as CO2 sources):** Recent studies in large river systems such as the Amazon and Congo have identified wetlands as previously unrecognized sources of CO2 and organic matter (Abril et al., 2014; Borges et al., 2015).

**L 42-46 (overestimation of pCO2):** While $pCO_2$ calculated from available water quality data such as pH and alkalinity has been used widely to estimate $CO_2$ emissions from a wide range of inland water systems (Lauerwald et al., 2013; Raymond et al., 2013), substantial overestimation of $pCO_2$ can occur in acidic, organic-rich inland waters due to the contribution of organic acids to alkalinity and the limited carbonate buffering (Abril et al., 2014).

L.43-45. I would suggest to add this reference where CO2, CH4 and N2O have been measured

simultaneously in the Zambezi River (Teodoru et al., 2015).

**<Response> The suggest reference has been cited in L 49, together with another recent paper reporting simultaneous measurements of three GHGs in a highly impacted river system (Borges et al., 2018).**

L.90-97. In my opinion, those sentences belong to the study site section.

**<Response> Yes, the sentences might fit into the study section. But we wanted to provide an overview of previous studies on anthropogenic perturbations to various reaches of the studied basin. Since this brief overview is different from detailed site descriptions in the following method section, we had to keep the overview section in the introduction.**

L.115-118. I would suggest to show land use on the map of the figure 1 (see my comments below for the figure 1).

**<Response> We could not show dominant land use types together on Fig., 1 because there are already too many symbols to show on Fig. 1. Therefore, we included an additional map showing 7 major land cover types as a supplementary figure (Fig. S1).**

[Figure]

L.135-136. According to Fig. 1, JN transect is an highly urbanized tributary but authors wrote that in this transect there is a forested headwater. This seems paradoxical to me (cf my comments of the Fig. 1).

**<Response> In addition to the additional site map (Fig. S1), more information about the land use of the urbanized tributary has been provided in L 131 (**~45% of which urban land use accounted for in 2014 (Seoul Metropolitan Government, 2017)**), along with an explanation of site selection based on land use in L 145-146 (**The 8 sites were selected to cover the spatial pattern of land use, ranging from the forested upper reach to the increasingly urbanized downstream reaches (Fig. S1).**).**

L.189. Please refer to Gran (1952).

**<Response> Cited.**

L.189. Usually, electro-titration of TA with the Gran method used 0.1N HCl as titrant.

**<Response> That's right. We have indicated 0.1 N HCl as the usual concentration used for the Gran method (L 199).**

L.204. Please insert period after "parameter".

**<Response> Inserted.**

L.218-220. There are two forested streams (one on the JN transect and one on the main transect, right?). To avoiding any confusion, I would suggest to specify between brackets the station name. Otherwise, the reader always needs to search this information in other figures or tables. I would suggest doing the same for the remainder of the text.

**<Response> Site names of two forested streams have been indicated throughout the manuscript.**

L.225-227. Visualizing the Fig.S1, I am not totally agree with author's comment. At the HR14 sampling station, N2O and CH4 seemed affected by season (notably spring and summer), as well at the HR2 and HR4 sampling stations where CH4 seemed affected by summer/winter seasons. In order to determine if seasons significantly affects GHG concentrations at a given station, I would suggest performing a Kruskall-Wallis test accompanied with a Dunn's test in order to accounting for the multiple comparison.

**<Response> The results of the suggested tests are indicated on Fig. S2. The sentence has been split and rephrased in L 239-242 (** $pCO_2$ tended to be higher in summer than in other seasons at all monthly monitoring sites except HR8 and HR 11, which are subject to direct or indirect influences of the cascade dams along the middle reach. There was no clear seasonality in $CH_4$ and $N_2O$ across the sites, but at the lower-reach site HR14 the concentrations of two gases tended to be higher in spring and summer than in fall and winter (Fig. S2).

L.228-237. I think that it would be interesting to know if decrease/increase described in this paragraph with the Figure 3 are statistically significant. For that, I would recommend performing a Mann-Whitney test between stations that are following each other's (testing HR1-HR2, then HR2-HR4…etc). In addition, I would suggest adding Mann-Whitney test results in the Figure 3.

**<Response> The results of Mann-Whitney U test have been added in Fig. 3, with their descriptions added in L 244-247 (** When Mann-Whitney $U$ tests were conducted to detect downstream changes between two successive sites, both DOC and FI were significantly different between two mainstem sites (HR11, HR14) and the urban tributary JN (HR 12). HIX generally decreased downstream along the river, with significant changes occurring during transitions from HR1 to HR2 and from HR4 to HR8 (Fig. 3).**).**

L.241-252. To understand basin-scale controls on CO2, CH4, N2O concentrations, authors explore their dataset by doing bivariate analysis (e.g., Kendall rank correlation) between either CO2, CH4 or N2O and each water quality parameter for the lower/middle/upper reach. This statistical test is appropriate but I think that a multivariate analysis (as PCA, may be associated with a cluster analysis of variable) with all parameter for each lower/middle/upper reach would be also very interesting. Another possible PCA would be a PCA biplot (graph of individuals and variables together), with all the dataset, in order to see where the lower/middle/upper reach points are situated with regards to the variability of the dataset. I supposed that a multivariate analysis will learn the authors more about the variability of the dataset, and how control patterns of CO2, CH4 or N2O evolved from upstream (upper reach) to downstream (lower reach). In addition, it will give information about which variables are important to describe the variability of the dataset. What do you think?

**<Response> In addition to the Kendall rank correlation, a PCA scatter plot has been included as a supplementary figure (Fig. S3). This plot supplements the cluster analysis suggested by the first reviewer (Fig. 4) and the Kendall analysis results (Fig. 5). Descriptions of this additional analysis have been provided in L 261-264 (** Reach-specific clustering of data was also found on a PCA scatter plot with two primary components accounting for 57.5% of variations (Fig. S3). While the upper and middle reach data were overlapped considerably on the PCA scatter plot (the upper reach with a wider scatter), the majority of the lower reach data were separated from the overlap of the upper and middle reaches.**).**

[Figure]

**Fig. S3.** Reach-based grouping of all measurments in the upper, middle, and lower reaches of the Han River alonng two components identified by principal component analysis (PCA).

L.276-278. Please add per mil symbols.

**<Response> Added.**

L.301. Richey et al (1988) is somewhat outdated, please add Abril et al. (2014).

**<Response> Added.**

L.304. "…regulated river system". May the authors add references?

**<Response> A relevant reference (Crawford et al., 2016) has been cited.**

L.304-308. This is a 6 lines sentence, quite difficult to follow, please consider revising the sentence.

**<Response>The sentence has been split and reformulated in L 329-334 (**It would be very challenging to tease out multiple, interrelated factors as shown by previous studies of GHG dynamics in urbanized river systems (Smith et al., 2017; Wang et al., 2017b). However, the observed longitudinal patterns of three GHGs (Figs. 2−4), along with their correlations with specific sets of water quality components (Fig. 5), make one thing clear: the primary factors and mechanisms for the production and consumption of three GHGs may change in response to longitudinal variations in dominant anthropogenic perturbations, often abruptly as shown by the localized pulses of GHGs downstream of urban tributary inflows (Figs. 2, 8).**).**

L.336. Please add references about methanotrophy in water column of lake (e.g., Morana et al., 2015; Roland et al., 2017).

**<Response> A relevant paper, together with descriptions of aerobic and anaerobic CH4 oxidation, has been included in L 361-362 (**aerobic and anaerobic $CH_4$ oxidation in water column with a depth-dependent gradient

of $O_2$ availability as a driving force for the observed spatial variations (Roland et al., 2017)**).**

L.343-346. In dam water column, you mentioned previously that the enrichment in CH4 originates from anaerobic conditions in organic-rich sediments. Usually, in strictly anaerobic conditions as occur for the methanogenesis, denitrification in the sediment is 'complete' producing N2 gas and not N2O. However, water column is oversaturated in N2O. How do you explain this? Did you measure GHG, O2 or NH4+/NO3- in the profile of the water column?

**<Response> No, we did not measure the depth profiles. The limited production of N2O under anaerobic conditions has been mentioned in L 371-372 (**although strictly anaerobic conditions might result in a more complete denitrification to $N_2$, contributing little to $N_2O$ production**).**

L.355-359. This is a 5 lines sentence, quite difficult to follow, please consider revising the sentence. In addition, it is not clear to me, all the data presented in this sentence originates from Yoon et al (2016)? Please, specify.

**<Response> The sentence has been split and reformulated (L 383-387:** When the estimated rates of $CO_2$ production, consumption, and outgassing along the downstream reach were compared in June 2016, the amount of $CO_2$ produced from organic matter biodegradation was much greater than the amount of $CO_2$ consumed by phytoplankton and similar to the $CO_2$ efflux to the atmosphere. In May 2015, when Chl *a* concentrations were much higher than in June 2016, the bulk of $CO_2$ delivered by the tributaries was estimated to be consumed by phytoplankton photosynthesis along the same reach.**).**

L.369.370. Authors mentioned that the amount of CH4 and N2O discharged from the WWTP appeared to drive the magnitude and temporal variability of the tributary inputs to the lower reach. When I observed the figure 6, this is necessary true for N2O, but not necessary true for CH4. Indeed, CH4 increased way before the appearance of the WWTP, and the two points of Nov 2015 and May 2016 that are very different suggest a high temporal variability that could explain CH4 concentrations measured at HR12. Do not you think that there is another source of CH4 than WWTP for this tributary?

**<Response> We agree that other upstream sources might also have influenced the observed large spatial and temporal variations of the tributary CH4. A sentence has been added in L 400-403 (**In the case of $CH_4$, however, the large spatial and temporal variations observed along the tributary upstream of the WWTP also point to the potential role of the benthic sediment as an upstream source of $CH_4$ (Stanley et al., 2016), although further research is needed to elucidate all important sources of the tributary $CH_4$.**).**

L.385.402. In this paragraph, could you explain the spatial longitudinal pattern of δ13C-DOC?

**<Response> An existing sentence has been split and added by two new sentences in L 422-427 (**In particular, large fluctuations in $\delta^{13}C_{DOC}$ along the upper to middle reaches from HR2 to HR11 do not present any consistent longitudinal trend of the stable C isotopic composition. However, distinct increases in $\delta^{13}C_{DOC}$ at the most downstream site (HR14) compared to the $\delta^{13}C_{DOC}$ at the forested headwater stream (HR1) indicate a potential contribution of autochthonous DOM components to the isotopic signature of the bulk riverine DOM, which deviated substantially from those of the headwater DOM dominated by allochthonous components (Fig. 7).**).**

L.388. Did you mean 72 among 695 or did you mean 72%? Please, specify.

**<Response> It has been clarified by adding % (72%).**

L.403. What does RKM term means? Please, specify?

**<Response> RKM has been replaced by "**km from the river mouth**" (L 439).**

L.403-416. All the statements you mentioned in this paragraph are maybe true but remain unclear to me. To improve this paragraph, I think that you need to better identify inputs and processes playing a role in the variability of δ13C-CO2 signature in the studied river. First, I am partially agree with the first sentence because dissolution of carbonates is CaCO3+CO2+H2O ⇌ 2HCO3- + Ca2+. Thus, dissolution of carbonates will not influence δ13C-CO2 signature but will influence δ13C-HCO3- and thus δ13C-DIC signature (e.g., Deirmendjian and Abril, 2018). Second, you mentioned δ13C-CO2 originating from riverine organic matter degradation. So, did you mean riverine organic matter coming from aquatic autotrophy? or riverine organic matter coming from soil and groundwaters leaching that is degraded in river? Because both sources have a distinct δ 13C-DOC signature. Thereafter you compared δ13C-CO2 value originating from riverine organic matter degradation with δ13C-CO2 value originating from lakes to highlight the fact that there are other processes than bacterial degradation to explain the variability δ13C-CO2 in your dataset. According to the δ13C-CO2 value of the lake, in this lake a high proportion of the CO2 originates from terrestrial degradation of DOC from C3 plants. So when you com your transect? In the third statement, you mentioned the preferential used of 12CO2 by heterotic bacteria, but how heterotrophic bacteria can used CO2? Please, clarify.

**<Response> We agree that there were some uncertainties in describing sources and processes related to the isotopic composition of riverine CO2. This is due to the fact that most studies have reported d13C in DIC, not in CO2. To respond to reviewer comments, we have clarified some unclear descriptions of DIC vs CO2 processes, as shown in the following revised paragraph (L 439-456):**

The longitudinal increase in $\delta^{13}C_{CO2}$ from −20.9‰ at 76 km from the river mouth to −16.7‰ at 50 km from the river mouth in Fig. 8 might be related to a complex array of interacting processes such as organic matter degradation, photosynthesis by phytoplankton, and atmospheric gas exchange, which have usually been investigated as determinants of the isotopic composition of riverine DIC consisting of dissolved $CO_2$, bicarbonate, and carbonate (Barth et al., 2003; Schulte et al., 2011; Zeng et al., 2011; Deirmendjian and Abril, 2018). The observed values of $\delta^{13}C_{CO2}$ fall within the reported ranges of $\delta^{13}C$ measured for $CO_2$ dissolved in riverine and estuarine waters (−25 – −15‰) (Longinelli and Edmond, 1983; Maher et al., 2013). However, the values reported here are less negative than the ranges of $\delta^{13}C$ measured directly for $CO_2$ respired by bacteria consuming organic matter of terrestrial and algal origin in two streams and eight lakes in Canada (−32.5 – −28.4‰) (McCallister and del Giorgio, 2008). When the observed values of $\delta^{13}C_{CO2}$ are compared with the low range of $\delta^{13}C_{CO2}$ reported by McCallister and del Giorgio (2008) and the usual ranges of $\delta^{13}C$ in plant and algal biomass as two primary biological sources of riverine $CO_2$ (Fig. 7), it follows then that other riverine processes than bacterial degradation of plant and algal biomass might be involved in the upward shift of $\delta^{13}C_{CO2}$. It has been reported that $\delta^{13}C$ in riverine DIC derived from carbonate dissolution and bacterial respiration ranges from −15 – −5‰, reflecting the balance between the concurrent processes that can either enrich or deplete DIC in $^{13}C$ (Telmer and Veizer et al., 1999; Barth et al., 2003; Schulte et al., 2011; Zeng et al., 2011). In contrast to the preferential use of the lighter organic C by heterotrophic bacteria depleting $^{13}C$ in the respired $CO_2$, photosynthesis and atmospheric gas exchange can result in an enrichment of $^{13}C$ in remaining riverine $CO_2$ through preferential phytoplankton uptake of the lighter $^{12}CO_2$ and dissolution of atmospheric $CO_2$ enriched in $^{13}C$, respectively (Schulte et al., 2011).

L.417-418. I supposed that you refer at the isotopic fractionation due to the thermodynamic equilibrium between CO2 and HCO3-? However, you cannot status only with this information that in your studied river δ13C-DIC signature will be 10‰ higher than δ13C-CO2 signature. Indeed, Equation of δ13C-DIC is δ13C-DIC= (δ13C-CO2* x [CO2*] + [HCO3-] x δ13C-HCO3- + [CO32-] x δ13C- CO32-) / ([CO2*] + TA) The signature of δ13C-DIC depends thus on complex interplays between initials concentration of each dissolved inorganic parameter as well as their signature, then processes producing or consuming DIC (primarily photosynthesis, degassing, respiration, weathering), and the isotopic thermodynamic equilibrium between each compounds.

**<Response> To reflect your concern, a caveat has been added in L 457-459 (**with a caution in mind that the actual $\delta^{13}C$ in DIC might be determined by various factors including initial concentrations and isotopic ratios of each DIC species and complex processes producing or consuming those DIC species (Deirmendjian and Abril,

2018)).

L.418.420. However, in the first part of the figure δ13C-CO2 increased at the same rate as in the second part of the figure but without any increase in Chl a. Please, explain.

**<Response> We have specified the reach where the general increasing pattern was observed in L 461-462** (general increases in Chl *a* along the lower reach flanked by two submerged weirs (69 – 50 km from the river mouth) (Fig. 8),**).**

L.421.422. Can you explain the difference in δ13C-CO2 between tributaries and main stem?

**<Response> Explanations for distinctive δ13C-CO2 in tributaries have been provided in L 465-469 (**The distinctively higher values of $\delta^{13}C$ observed for the tributary $CO_2$ might have resulted from a combination of processes, including the same photosynthesis and atmospheric gas exchange as occurring in the mainstem and tributary-specific processes such as the transport and transformations of anthropogenic organic matter in urban wastewater. WWTP effluents have been shown to contain old organic matter with characteristic C isotopic composition (Griffith et al., 2009; Griffith and Raymond, 2011; Butman et al., 2015).

L.431.432. Does degassing of CH4 to the atmosphere could have an impact on the upstream-downstream decrease of CH4?

**<Response> Loss of CH4 through evasion has been mentioned in L 480 (**and/or evasion of $CH_4$ to the atmosphere**).**

L.445. To conclude, do you have any recommendations for politician, river managers and stakeholders to improve water quality and reducing GHG concentrations in highly urbanized watershed?

**<Response> A concluding remark on integrated river basin management has been added in L 532-535 (**Identifying hot spots of water pollution and GHG emissions in highly human-impacted river systems would contribute to establishing novel river basin management options integrating the traditional water quality control and an emerging challenge of climate change mitigation by helping watershed managers set priority areas of policy responses to multiple concurrent environmental stresses.**).**

Tab. 1. I would suggest adding a left column to specify upper/middle/lower reach.

**<Response> A column has been included in Table 1.**

Fig. 1: I am not aware if a land use database exists for South Korea, but if such a database exists, I would recommend adding the land use in the map of the Figure 1, particularly to visualize where croplands, forest and cities are located. In addition, to visualize the proportion of croplands, forest and cities in the studied catchment. I would also suggest adding the forested headwater from the JN transect in another color than the other points of the JN transect. Indeed, JN transect is considered by the authors as an urban transect, and thus, this is strange to associate an urbanized river with a forested headwater. Perhaps, authors could also apply a different typology for the sampling points, with for example, one color for forested streams, one for agricultural streams…It would be easier to visualize sampling points in the map of the Figure 1. Please, also add metric scale on the map.

**<Response> As explained before, we have prepared an additional map showing land use (Fig. S1).**

Fig. 2. I would suggest specifying upper/middle/lower reach in the figure, perhaps at the top of the figure.

**<Response> Three reaches have been specified at the top of the figure.**

Fig. 3. I would suggest specifying upper/middle/lower reach in the figure, perhaps at the top of the figure.

**<Response> Three reaches have been specified at the top of the figure.**

Fig.4. I would suggest to specify which tributaries belong to the red points (JN? TC? AN? or this is just HR12?)

**<Response> Tributaries have been specified in the caption of a new figure made following the first reviewer's suggestion.**

Fig.8. I would suggest to specify sampling stations names on the graphs.

**<Response> Three mainstem sites have been marked on the graphs.**

---

## Author Comment (AC3) · 19 Aug 2018

Thanks for providing us with detailed information about may references. Relevant references have been cited in the revised manuscript, as detailed in our response to RC2.

---

## Author Comment (AC4) · 19 Aug 2018

**bg-2018-278**

**Author response to RC4 (3[rd] reviewer's comments)**

Jin and co-authors present an extensive dataset of greenhouse gas (GHG) measurements along a human-impacted river in Korea. The river is divided in three sections: the upper reach which is characterised by forest and agricultural land use, the middle reach which is impacted by multi-purpose dams and the lower reach which is influenced by wastewater discharge of the city of Seoul. Significant discontinuities in the GHG concentrations were found in the dam and sewage impacted reaches. Although the conclusions are not very surprising, the importance of this manuscript is the comprehensive dataset created by the authors, which provides a lot of quantitative information for larger-scale overview articles.

General comments

In the introduction, you often mention that previous studies looked at only a single anthropogenic factor. It took me a second reading before I distinguished the two anthropogenic factors, dams and sewage, as spatially distinct along the river (middle and lower reach). Even though it might be a slight over-simplification, it might help the reader if you make it more explicit (similar to the second sentence of previous paragraph). Your many sites and tributaries can become confusing, but framing it as 'natural', 'dams' and 'urban/sewage' would help to keep track.

**<Response> As you indicated, the suggested framing is difficult to apply considering within-reach spatial heterogeneity. To follow your suggestion, we have added some additional sentences to specify the dominant anthropogenic perturbation of each reach in Introduction (Lines 104-105:** The primary objective was to examine the effects of dams and urban wastewater**) and Methods (L 138-141:** Compared to the upper reach (HR1 – HR4) located in a heavily forested watershed with some scattered agricultural areas, the impounded middle reach (HR5 – HR11) and the lower reach receiving heavy loads of urban sewage (HR12 – HR15) are subject to stronger anthropogenic perturbations; **L 145-146:** The 8 sites were selected to cover the spatial pattern of land use, ranging from the forested upper reach to the increasingly urbanized downstream reaches (Fig. S1).**).**

You have a tendency to make complicated sentences because you want to include all your reasoning or justifications in one sentence. While these sentences were grammatically correct, they are really hard to read. Be critical to sentences which are more than 4 lines and consider splitting them up. I will indicate a few of those sentences in the detailed comments.

**<Response> We have reformulated the long sentences you pointed out, as detailed in our responses to specific comments below. In addition, we have thoroughly revised many parts of the manuscript to improve the readability by minimizing long sentences.**

Specific comments

L. 16: I have difficulties with calling the dams and sewage primary controls, because I perceive the term 'primary' as the 'first', while the human impact is actually superimposed on the natural dynamics. I would suggests changing it to "major controls". Also, the effects are not the controlling the GHG dynamics. "... to investigate the influence of dams and urban water pollution on GHG dynamics ..."

**<Response> The phrase has been changed to "**to investigate dams and urban water pollution as major controls**" in L 16.**

L. 28 : might (without e at the end)

**<Response> Corrected.**

L. 112: Add the length of the river

**<Response> The length, together with a reference, has been added in L 116-117.**

L. 115-118: Split over two sentences. One about major land use, one about the metropolitan area.

**<Response> Split into two sentences (L 119-122:** Major land uses in the basin include forests (73.6%), croplands (14.1%), urban and industrial areas (2.6%), and other uses (9.7%) (Fig. S1). The highly urbanized metropolitan area along the lower reach has a large impermeable surface regarded as urban land use, accounting for 58% of the total city area of Seoul (Seoul Metropolitan Government, 2017).**).**

L.127: What is the treatment level of the three WWTPs.

**<Response> Information available for the largest WWTP has been provided in L 133-134 (**which employs tertiary treatments including modified Ludzack Ettinger (MLE) and anaerobic-anoxic/oxic process ($A_2O$))**.**

L. 138-140: What are the observation dates/month & year?

**<Response> DMY (**10 June 2016**) has been added in L 149.**

L. 219: It was not clear to me where the agricultural stream and forested headwater stream belong to. Are they both part of the upper reach? Also the submerged weirs is not clear to which section they belong. It felt like you are jumping up and down along the river in the description of the longitudinal variations. Try to be consistent in describing each parameter from upper reach over middle reach to lower reach.

**<Response> The paragraph has been rearranged and added with site information so that the longitudinal variations are described in the order of upper-middle- lower reaches.**

**L 230-239:** The concentrations of three GHGs were relatively low along the upper reach, although small, but noticeable increases occurred in the agricultural stream (HR2) compared to the generally low values found in the forested headwater stream (HR1) (Fig. 2; Table 1). Levels of $p$CO$_2$ in the middle reach (HR5 – HR11; 51 – 761 µatm) tended be lowest when compared with upper and lower reaches and were particularly low at sites within a few km upstream or downstream of the cascade dams. In contrast, N$_2$O and CH$_4$ concentrations were higher at one (HR6; 212 nM N$_2$O L$^{-1}$) or three dam sites (HR6, HR7, and HR10; 693 – 748 nM CH$_4$ L$^{-1}$), respectively, compared to the upstream or downstream reaches of the dam sites (Table 1). For all three GHGs, large downstream increases were found along the lower reach flanked by two submerged weirs (HR12 – HR14). Gas concentrations at some lower-reach sites approached or exceeded the levels found in three tributaries draining the urban sub-catchments located in Seoul and surrounding suburban areas (Fig. 2).

L. 225: replace "less impacted upstream or downstream reaches" with "compared to the upper and lower reaches". All of the reaches are impacted, just in a different way.

**<Response> Here upstream and downstream do not refer to upper and lower, respectively. They literally mean upstream and downstream reaches of the dam sites. The phrase has been changed to "**compared to the upstream or downstream reaches of the dam sites**" (L 236).**

L. 225-227: This is a complicated sentence. Consider splitting it up (especially the explanation for sites HR8 and HR11).

**<Response> The sentence has been split and rephrased in L 239-242** ($pCO_2$ tended to be higher in summer than in other seasons at all monthly monitoring sites except HR8 and HR 11, which are subject to direct or indirect influences of the cascade dams along the middle reach. There was no clear seasonality in $CH_4$ and $N_2O$ across the sites, but at the lower-reach site HR14 the concentrations of two gases tended to be higher in spring and summer than in fall and winter (Fig. S2).**).**

L. 233: What is the water discharge ratio between the tributary and the main river?

**<Response> Discharge ratios have been provided in the rephrased sentence in L 249-253** (The comparison of monthly water quality measurements between the six sites and the urban tributary (HR12), together with the proportion of tributary discharge in the mainstem flow ranging from 5% in the monsoon period to 12% in dry seasons, points to the disproportionate influence of urban tributary inputs on the downstream increases in concentrations of DOC and nutrients observed in the lower reach (Fig. 3).**).**

L. 238: This is a complicated sentence. "When we pooled the measurements for the whole river basin, at least two of the GHG's exhibited significant ..."

**<Response> The sentence has been rephrased in L 257** (When all measurements of three GHGs and water quality were pooled for the whole river basin,…**).**

L. 260: How can the WTTP effluents and tributary reach values of the upstream river. Consider rephrasing.

**<Response> The sentence has been rephrased in L 283-285** (In contrast, $CH_4$ concentrations exhibited relatively large fluctuations along the middle reach, ending up at the intermediate levels observed for the upper to middle reaches in the WWTP effluents and the tributary outlet.**).**

L. 261: the large scatter (without s)

**<Response> Corrected.**

L. 273: "though" doesn't seem the correct word.

**<Response> Corrected** ("through"**).**

L. 304-309: Very long and complicated sentence with lots of subsentences.

**<Response> Reformulated in L 329-334** (It would be very challenging to tease out multiple, interrelated factors as shown by previous studies of GHG dynamics in urbanized river systems (Smith et al., 2017; Wang et al., 2017b). However, the observed longitudinal patterns of three GHGs (Figs. 2−4), along with their correlations with specific sets of water quality components (Fig. 5), make one thing clear. The primary factors and mechanisms for the production and consumption of three GHGs may change in response to longitudinal variations in dominant anthropogenic perturbations, often abruptly as shown by the localized pulses of GHGs downstream of urban tributary inflows (Figs. 2, 8)**).**

L. 408: Could the composition of the respired organic material be responsible for the variation in δ13C? I expect

very little C4 plants in Canada, which is consistent with the very low δ13C values. If you have more variation in C3-C4 plants throughout your catchment, then you would expect to see that change reflected in the riverine C.

**<Response> We understand your point, but the lower reach is in the Seoul metropolitan area with little agricultural area, suggesting that variations in C3/C4 plants cannot explain spatial variations in δ13C in CO2 along the lower reach mig. To provide more coherent explanations, the entire paragraph has been revised as follows (L 439-456):**

"The longitudinal increase in $\delta^{13}C_{CO2}$ from −20.9‰ at 76 km from the river mouth to −16.7‰ at 50 km from the river mouth in Fig. 8 might be related to a complex array of interacting processes such as organic matter degradation, photosynthesis by phytoplankton, and atmospheric gas exchange, which have usually been investigated as determinants of the isotopic composition of riverine DIC consisting of dissolved $CO_2$, bicarbonate, and carbonate (Barth et al., 2003; Schulte et al., 2011; Zeng et al., 2011; Deirmendjian and Abril, 2018). The observed values of $\delta^{13}C_{CO2}$ fall within the reported ranges of $\delta^{13}C$ measured for $CO_2$ dissolved in riverine and estuarine waters (−25 – −15‰) (Longinelli and Edmond, 1983; Maher et al., 2013). However, the values reported here are less negative than the ranges of $\delta^{13}C$ measured directly for $CO_2$ respired by bacteria consuming organic matter of terrestrial and algal origin in two streams and eight lakes in Canada (−32.5 – −28.4‰) (McCallister and del Giorgio, 2008).When the observed values of $\delta^{13}C_{CO2}$ are compared with the low range of $\delta^{13}C_{CO2}$ reported by McCallister and del Giorgio (2008) and the usual ranges of $\delta^{13}C$ in plant and algal biomass as two primary biological sources of riverine $CO_2$ (Fig. 7), it follows then that other riverine processes than bacterial degradation of plant (predominantly C3 in the studied basin) and algal biomass might be involved in the upward shift of $\delta^{13}C_{CO2}$. It has been reported that $\delta^{13}C$ in riverine DIC derived from carbonate dissolution and bacterial respiration ranges from −15 – −5‰, reflecting the balance between the concurrent processes that can either enrich or deplete DIC in $^{13}C$ (Telmer and Veizer et al., 1999; Barth et al., 2003; Schulte et al., 2011; Zeng et al., 2011). In contrast to the preferential use of the lighter organic C by heterotrophic bacteria depleting $^{13}C$ in the respired $CO_2$, photosynthesis and atmospheric gas exchange can result in an enrichment of $^{13}C$ in remaining riverine $CO_2$ through preferential phytoplankton uptake of the lighter $^{12}CO_2$ and dissolution of atmospheric $CO_2$ enriched in $^{13}C$, respectively (Schulte et al., 2011)."

Figure 2: Could you indicate the three different reaches in the graphs?

**<Response> Three reaches have been indicated at the top of the graphs.**

---

## Author Response (AR2)

EWHA WOMANS UNIVERSITY

Ji-Hyung Park
Professor
Ecosystem/Biogeochemistry Lab
Dept. Environmental Science & Engineering          Telephone: +82-2-3277-2833
Ewha Womans University                             Fax: +82-02-3277-3275
Seoul 10375, Republic of Korea                     E-mail: jhp@ewha.ac.kr
                                                   Homepage: http://peblab.com

Editor
Biogeosciences                                                    October 10, 2018

Dear Dr. Abril,

We appreciate your detailed and insightful comments on the revised manuscript.

To respond to your comments and suggestions, we have thoroughly revised our manuscript and included some new data such as alkalinity and calculated $\delta^{13}$C-DIC values, as detailed in the enclosed list of detailed responses. Many additional changes made in this version have been marked by a weak blue color to distinguish them from the blue-marked previous changes in the first revision.

We would appreciate your kind consideration of our revised manuscript for publication in Biogeosciences.

Sincerely,

Ji-Hyung Park

Cc: Hyojin Jin, Tae Kyung Yoon, Most Shirina Begum, Eun-Ju Lee, Neung-Hwan Oh, Namgoo Kang

**<Responses to Editor's comments on the first revision>**

Because the first round of reviews of your paper was quiet detailed and comprehensive, I wanted to move forward its edition process based on my own reading. My conclusion is that your paper contains all the necessary data to reach your conclusions, that these conclusions are original and well within the scope of Biogeosciences, but that presentation and selection of the data must re-thought, and the text must be intensively re-worked in order to make your conclusion clear and understandable for all BG readers. Note that my evaluation is based only on text and figure and not sup. Material

One first big problem with your MS is your phrasing that often creates confusion. On many occasions, what should be a strong and precise message is diluted in very general statements using sophisticated expression likely of your own invention. In other occasions, the structures of sentences are very long and repetitive and something that could be clearly said in a few words is hardly expressed in a long and awkward sentence. The style of your text is really a major weakness of your paper because it dilutes the main messages in unnecessary superficial text, including the occurrence of some wrong or imprecise statements. Here an example: L500 "The observed reach-specific patterns of altered water quality and GHG dynamics provide an empirical evidence for ecosystem structural and functional responses to anthropogenic changes in hydrogeomorphic patches of the fluvial landscape, which have been emphasized in recent conceptual models integrating fluvial geomorphology and ecosystem processes at the valley to reach scales (Thorp et al., 2010)." This looks like a suite of savant words without real precise meaning and without referring to objective scientific facts to sustain the statements: what is a "reach-specific pattern"? What pattern? of increasing/decreasing what parameter? "water quality" and "dynamics" are very general terms that can be either positive of negative; "ecosystem structural and functional responses" ecosystem structure is a vast domain, be more precise : what ecological processes are your dealing with here? What responses? What is a "hydrogeomorphic patch"? sorry, I do not understand and I doubt this expression has been defined in the literature. What is a "reach scale"? how many km for the reach scale?… etc., etc. You use the term "reach" (upper reach, lower reach, river reach… etc.) throughout your MS without giving a clear definition for it. Other kind of problem with your style is illustrated by the following sentence from the abstract (there are many other cases in the MS) "The basin-wide surveys of three GHGs revealed distinct increases in the concentrations of three gases along the lower reach receiving urban tributaries enriched in GHGs and DOC". Your are repeating 3 times "GHG" here and 2 times "three GHGs" although the meaning of the sentence is simply that urban tributaries are a source of GHG and DOC for the river. This kind of phrasing occurs all through your MS and makes it very hard to read and follow, because crucial new original finding are diluted in very general statements, without clearly hierarchizing what is new, what is specific from the study site, what can be generalized elsewhere, and, where and when a statement applies… You will find a list of problematic sentences below, because I am not English native, the list is probably not exhaustive, and I did not try to re-phrase all of them. Please consider seriously that your MS needs profound revision of the style, including detailed editing of language meaning, avoiding vague conceptual statements not based on precise scientific facts (eg "the parameter X increases downstream when the parameter Y decreases" and not "parameters X and Y evolve according to complex biogeochemical processes occurring from upper to lower reaches of river basin and tributaries…"). Simplifying and shortening sentences so it becomes easier for readers.

**<Response> We agree that there were many long and general descriptions in the previous version. To avoid any unnecessary confusion, we have tried our best to rephrase all the long, general, or redundant descriptions. In the case of Thorp et al citation, we used the terms and descriptions as appeared in the paper to respond to a reviewer's suggestion. We understand your point, so the sentence has been removed from the text. The rewritten paragraph does not require this sentence any more. "Reach" is now used based on our definitions of the upper, middle, and lower reaches provided at their first use in the abstract and main text. Please note that we have revised the entire manuscript including the sentences you mentioned here and below.**

Second general problem is the choice of data you are presenting in the MS. The MS deals preliminary with CO2, CH4 and N2O (or GHGs); Ancillary parameters include, nutrients, oxygen, pH, CHla, alkalinity, DIC DOC concentration, 13C and 14C, UV absorbance, and fluorescence excitation. I found questionable the choice you made in showing these data in the figures and tables and in the supplementary material.

UV absorbance, and fluorescence excitation bring very little information to the paper, I wonder if these data are really necessary here (they could appear in a couple of sentences in the discussion as "unpublished data" avoiding long description in the Mat &Met.

**<Response> Our manuscript is not just about longitudinal patterns of the three GHGs. We also need to provide explanations of the key controls. In this sense, DOM optical properties are invaluable to characterize anthropogenic DOM abundant in the lower Han River in relation to GHG dynamics. To make this point clear, we have added key findings on DOM properties and their implications for longitudinal patters of both DOM and GHGs in the abstract and results/discussion.**

To the contrary, because your data include pCO2, and because pCO2 was measured with a headspace technique, the information on the entire carbonate system is necessary: provide TA values and calculated DIC values (from TA and pCO2). Values of these crucial parameters are not even mentioned in the text. WWTP generally release DIC in the form of TA, it should be the case here, a detailed analysis of TA might be important for the paper.

In addition, if TA values are high, the buffer capacity of the carbonate system should be taken into consideration in order to calculated in situ pCO2 from pCO2 measured in a headspace. The henry's law is not sufficient in the case of carbonate rich waters, because equilibrium between CO3= and HCO3- and between CO2 and HCO3- are displaced when creating the headspace (CO2 decreases, CO3= increases and HCO3- may increase or decrease depending on the conditions). Thus for alkaline waters, calculation of in-situ (pre-headspace) pCO2 must include the entire DIC and the changes in CO3=, HCO3- and CO2. In the headspace technique, the volume of water is finite; this is different from the equilibrator technique, where an infinite volume of water is equilibrated with a finite volume of air.

**<Response> We have included TA data (Tables S1-S3, S5, S7). We also understand the importance of TA and DIC in addressing pCO2 dynamics, but we could not include this topic for two reasons. First, another manuscript is now being prepared to compare measured pCO2 and calculated DIC species. Second, the key goal of the current manuscript is to compare the three GHGs measured using the same headspace equilibration technique. Because we deal with the multiple gases and numerous ancillary data, we thought that the topic of DIC should be addressed in a devoted manuscript to maintain our focus on three-gas comparison. Please understand that our approach is not different from other papers addressing multiple gases. And that this manuscript is already overloaded with different sets of data and topics.**

One of the reviewers pointed out the question of d13C-CO2 and I agree with her/him that stable isotopes studies must rely on data of d13C-DIC and not d13C-CO2. You mention in your revised MS a difference of about 10 to 11 per mil between d13C-DIC and d13C-CO2, referring to Mc Callister and del Giorgio. I haven't read this paper, but almost all the literature reports d13C-DIC and not d13C-CO2. The problem is that the difference between d13C-DIC and d13C-CO2 and the value d13C-CO2 are strongly dependent on water temperature, pH, and alkalinity, because the fractionation between CO3=, HCO3- and CO2 depend on these parameters. To the contrary, d13C-DIC is a conservative notion that considers all the 12C and the 13C contained in the DIC of a sample, whatever the temperature and/or the pH and whatever the proportion of each chemical form.

**<Response> We have cited more papers employing d13C-CO2 measurements (e.g., Campeau et al., 2017) and provided more detailed interpretations of our measurements in comparison with other papers. We also calculated d13C-DIC according to your suggestion, and compared with more literature information on d13C-DIC to describe implications of our findings. Please note that the entire section has been almost rewritten (L 460-563).**

You put a lot of emphasis in your MS on the importance and interest of using the dual isotopes approach; however, you apply this approach only to DOC in a very limited amount of samples. And in fact the dual isotope approach brings little information because of the contamination of WWTP with old DOM such as gasoline. If the paper deals mainly with GHG, one would expect the dual isotopes techniques to be applied to DIC or CH4... So be less ambitious when introducing the dual isotope approach because when applied to DOC here, it gives little information for the interpretation of the three gases

**<Response> We agree that our dual isotope approach is limited in many aspects. First of all, the small number of samples has been indicated in the abstract. We also paid more attention to interpreting the dual isotope ratios of DOM and tried to restrict our data interpretation to source**

**tracking. For example, we focused more on wastewater-derived aged DOM and gases when we linked DOM isotope data to gas isotope ratios, as detailed in the thoroughly revised section 4.2.**

Inversely, discussion on 13C-CH4 data is relatively superficial and could be strengthened and be more quantitative. These values are quiet high and reveal strong oxidation, maybe up to 95% of the CH4 is oxidized
**<Response> We have provided more explanations for CH4 oxidation based on additional references and observed values in the revised paragraph (L 540-563).**

Alternative choices of data to be shown in figure and table in the main MS are necessary, reminding that the main topics is CO2, CH4 and N2O
**<Response> Explained.**

Below the detailed comments on the MS

Repeating "GHG" >8 times in the abstract is awkward. Instead, specify at least the ranges of concentrations of each individual dissolved gas. Some key numerical values are missing in the abstract.
**<Response> The entire abstract has been rewritten to reduce redundant terms and expressions and provide key values.**

L35 not sure "body of research" is appropriate here
**<Response> The whole sentence has been changed to "**A growing number of studies have provided a wide range of estimates for the global greenhouse gas (GHG) emissions from inland waters**"**

L40 specify that wetland are source of dissolved CO2 FOR RIVERS (not source of CO2 for the atmosphere as it suggests here)
**<Response> The sentences now reads "**Recent studies in large river systems such as the Amazon and Congo have identified wetlands as previously unrecognized sources of $CO_2$ and organic matter for rivers**".**

L45 I guess you mean Abril et al. 2015 and not 2014
**<Response> Corrected.**

L49 change "natural" to "pristine"
**<Response> Changed.**

L53 what's the meaning of "concurrent" here? Please rephrase
**<Response> The sentence has been changed to "**While global river systems are now subject to multiple environmental stresses, including water pollution, impoundments, and climate change, most research efforts have addressed these multiple stresses separately.**".**

L55 insert measured "simultaneously" CO2, CH4 and N2O and remove "together"
**<Response> Done.**

L56 what's the meaning of "interrelated"? Please rephrase
**<Response> The phrase has been changed** (some common longitudinal patterns of gas concentrations determined by major sources and production mechanisms**).**

L60 "comparison of three GHGs" do you mean "these three GHGs" or could be any other GHG?
**<Response> The word "**these**" has been added.**

L70 "measurements of multiple GHGs…" you are repeating what you already stated before
**<Response> The sentence has bee rephrased** (Several recent studies conducted in highly human-impacted river systems have found unique longitudinal and seasonal patterns of $CO_2$ and other GHGs that might be explained by different factors and mechanisms from those relevant to large pristine rivers**).**

L83 "shifting balance between autotrophy and heterotrophy at diel to decadal scale" please be precise: shifting in favour of what? Favouring autotrophy or heterotrophy? All through the MS avoid the numerous sentence with such incomplete information.
**<Response> The sentence has been rewritten (**GHG dynamics in impounded waters and sediments may be explained by temporal changes in a suite of concomitant metabolic processes including primary production, methanogenesis, methane oxidation, nitrification, and denitrification**).**

L85 what is a "reach" ? I understand it is a portion of a river, but what is its typical length? Few meters, hundred meters, kilometres? The continuous use of this term all through the MS is really perturbing.
**<Response> The term in L85 has been changed to "**impounded waters and sediments**". When we use the term "reach" to indicate the three compared river sections (upper, middle, and lower), it is based on the definitions provided at its first use in the abstract (L 16-) and main text (L 108-). Otherwise, it is used to indicate a river section with similar structure and function, as commonly used in the literature. It should be noted that there is no agreed length definition. For example, hydrogeomorphologists often deal with short reaches < km (e.g., Poole, 2002), but some river biogeochemists use very long reaches > 1000 km (e.g., Richey et al., 1988).**

Although you repeated several times the necessity of measuring the three GHGs, nothing on the interests of measuring isotopes in the abstract.
**<Response> As described earlier, we have restricted the interpretations of isotope data, as described in L 31-36.**

L110 "we expected that the comparison of reach-specific spatial pattern of three GHGs and C isotopic composition in DOM, $CO_2$ and $CH_4$ … emerging concepts of anthropogenically created discontinuities in riverine metabolic processes and GHG emissions". Long, awkward sentence. What is a "reach-specific spatial pattern"? Do you simply mean that the longitudinal (or spatial) distributions of dissolved gases are probably impacted by dams and urban areas? "metabolic processes" do you mean primary production and/or respiration? Or other food-web metabolic processes?
If later in the MS you refer to the river continuum concept, it would be interesting to mention it in the intro.
**<Response> The sentence has been rephrased to be more specific, though it is still long (**The comparison of the three GHGs, DOC, and other ancillary water quality measurements across the three reaches affected by different anthropogenic perturbations would provide empirical data that can be incorporated into the emerging concept of anthropogenic discontinuities in riverine metabolic processes involved in primary production, organic matter degradation, and GHG emissions.**). The river continuum concept has been introduced in a preceding paragraph (L 72-76).**

L150 change "reported" by "described"
**<Response> Changed.**

L151 "in each SAMPLING, water SAMPLE…" rephrase
**<Response> "In each sampling" has been removed.**

L153: we don't care about the brand of the peristaltic pump
**<Response> The brand name has been removed, though some people cared in other review processes.**

L157 we don't care about the brand of the syringe, exetainer (L160), temperature and pressure sensor (L164)
**<Response> All those brand names have been removed.**

L178 Your analysis of d13C in $CO_2$ in a headspace without acidification is problematic. Instead, measurements of 13C-DIC should have been made.
**<Response> We are also aware of the usefulness of d13DIC analysis, but had a different objective, namely to track downstream changes in d13C-CO2/CH4, as a growing number of studies have used CRDS measurements. We hope that the rewritten section 4.2 would shed some insight on gas isotopic ratios.**

L183-184 "The concentration of total suspended solid (TSS) was measured as the difference in the filter weight before and after drying at 60°C for 48 hours." this procedure will give you the water content in the filter and filtered material, not the TSS.
Did co-authors did not revise the MS?
**<Response> Rephrased** (The concentration of total suspended solid (TSS) was determined by filtering a known volume of water sample through a pre-weighed GF/F filter and then weighing the filter again after drying at 60°C for 48 hours**).**

L204: "subsamples of filtered water samples" please reword
**<Response> Rephrased** (Some filtered water samples**).**

You do not explain how d13C-CO2 and d13C-CH4 were measured: GC/C/IRMS?
**<Response> Already described in a preceding paragraph (…**analysed for stable C isotope ratios of $CO_2$ ($\delta^{13}C_{CO2}$) and $CH_4$ ($\delta^{13}C_{CH4}$) by a GasBench-IRMS (ThermoScientific, Bremen, Germany) at the UC Davis Stable Isotope Facility.**).**

L234 "tended to be lowest" please reword
**<Response> The entire sentence has been rewritten, beginning now with "**The $pCO_2$ values at the four dam sites, ranging from 51–761 µatm, averaged 304 µatm, lower than the level expected for atmospheric equilibrium (~ 435 µatm).**"**

L234 "particularly low: provide value"
**<Response> The ranges of pCO2 values have been provided in L 242-247.**

L243 "of two gases" > of these two gases
**<Response> The word "these" has been added.**

L251 "points to" check English language
**<Response> The whole sentence has been reformulated** (Given the relatively small proportion of tributary discharge in the mainstem flow ranging from ~5% in the monsoon period to 12% in dry seasons, the comparison of monthly water quality measurements between the six sites and the urban tributary (HR12) illustrates the disproportionate influence..**)**

L252 "fraction" is a more usual word than "moieties"
**<Response> The term "moieties" has been changed to a common term used in the literature (protein-like DOM "components").**

L254 "three nutrients" which ones?
**<Response> Specified (**three major nutrients ($NH_4^+$, $NO_3^-$, and $PO_4^{3-}$)**).**

Nomenclature HR11… HR14… etc. are difficult to follow, indicating the characteristics of the river sections would help
**<Response> Site characteristics have been added to the site names** (from HR11 downstream of the last cascade dam to HR14 in the middle of the lower reach**).**

L260… you mention correlations with parameters such as DO, pH, TA but the numerical values of these important parameters only appear in the supplementary material. Maybe an additional table could help
**<Response> Please note that Fig. 5 can provide an overview of very complex correlations between GHGs and ancillary measurements. We still hope that readers can refer to more detailed numerical information provided in supplementary table (Table S4).**

L279 "pointed to WWTP effluents driving the concentrations of…" revise English language
**<Response> Rephrased** (revealed the dominant influence of WWTP effluents**).**

L280-283 you are repeating twice "GHGs" and "WWTP" in the same sentence, awkward sentence

**<Response> The sentence has been reformulated (**All the three GHGs exhibited similar levels and variations in the WWTP effluents and the tributary outlet, indicating a strong contribution of treated wastewater to the tributary gas export to the lower Han River.**).**

L283 "increase abruptly along the most downstream reach after passing the WWTP located within a few km upstream" do you mean they " rapidly increase immediately downstream of the WWTP"?
**<Response> Changed (**Both $p$CO$_2$ and N$_2$O concentrations in the tributary abruptly increased along the terminal section downstream of the WWTP**).**

L284 what does "ending-up" mean here? End of what?
**<Response> The whole sentence has been rewritten (**In contrast, CH$_4$ concentrations were very low at the three upstream sites, exhibited large fluctuations along the middle reach, and decreased slightly in the terminal section downstream of the WWTP.**).**

L285 "Corresponding to the large scatter of the box plots representing three GHG concentrations measured at the WWTP effluents and outlet site, two locations exhibited similar patterns of temporal variations in GHG concentrations" awkward and confusing sentence, please rephrase.
**<Response> Rephrased (**The levels of the three gases displayed large, similar temporal variations at the WWTP effluents and the tributary outlet site (Fig. 6e, 6f, 6g).**).**

L290-298 specify if "high" and low" delta14C values correspond to "old" or "young" DOC
**<Response> Specific delta 14 C values and corresponding ages have been added in L 319-.**

L296 unit is missing after -100
**<Response> Added.**

L300 "Concentrations of three GHGs (Fig. 8; Table S7) combined with d13C in CO2 and CH4 (Fig. 8; Table S8) collected along the lower Han River during a cruise expedition revealed clear tributary effects on the C isotopic composition of two GHGs sampled at the mainstem site" very awkward sentence. In addition it says almost nothing in terms of "results"
**<Response> The revised sentence reads "**The concentrations of the three GHGs and the values of $\delta^{13}$C in CO$_2$ and CH$_4$ (Fig. 8; Tables S7, S8) measured along a cruise transect exhibited large increases in gas concentrations and either gradual increases in $\delta^{13}$C$_{CO2}$ or abrupt decreases in $\delta^{13}$C$_{CH4}$ along the confluence of the urban tributary (HR12).**".**

L303 "d13CO2 continued to increase toward the river mouth, with its values bracketed by those measured for the two upstream tributaries and a downstream tributary" to what figure does this refer to? Is "values bracketed by" correct English?
We would need d13C-DIC and DIC concentrations here
**<Response> The new sentence now reads "**The gradual downstream increases in $\delta^{13}$C$_{CO2}$ along the mainstem transect reflected the tributary contributions to the mainstem isotopic composition, because the values found in the two upstream tributaries (−18.2‰, −18.3‰) and a downstream tributary (−14.7‰) were higher than the upstream mainstem values (Table S8).**". Regarding d13DIC, please refer to our response to a major comment on the same issue.**

L308 remove "three"
**<Response> To be consistent, we use "the three GHGs".**

L328 "lack of these natural source" do you mean the absence of floodplain? Not sure Richey 1988 considered wetlands as the "primary factor".
**<Response> The sentence has been rewritten (**Those natural sources are rarely found in the Han River basin, where the middle and lower reaches have been modified substantially by man-made structures. This lack of natural sources, combined with the differential patterns of the three GHGs attributed to dams and urban wastewater, suggests that increased water retention time and nutrient enrichment may play crucial roles in the production and consumption of the three GHGs in this highly regulated river system (Crawford et al., 2016)**).**

L332 and at many other places: not sure "of three GHGs" is correct (it could be other gases), I would say either "the three GHGs" or "these three GHGs"
**<Response> Corrected throughout the manuscript.**

L335. "Spatial variations in three GHGs observed along the middle reach (Figs. 2 □ 4) suggest that complex interacting metabolic processes in water column and sediment influence the levels of three GHGs to varying degrees depending on gas and reservoir"
Avoid this type of sentences that says basically nothing. What "metabolic processes" ? be precise. "depending on gas and reservoir" Awkward wording. Again the use or "three GHGs" is confusing
**<Response> Rewritten (**The highly variable concentrations of $CH_4$ and $N_2O$ along the middle reach in contrast to the consistently low levels of $pCO_2$ at the dam sites (Figs. 2 – 4) suggest that the rates of concomitant metabolic processes involved in the production and consumption of these gases in reservoir water and sediments may vary with predominant dam conditions such as water depth and sediment accumulation.**).**

"Lower values of pCO2 measured at all impoundment-affected sites including site HR11 downstream of the last dam (HR10) indicate an enhanced planktonic CO2 uptake, in agreement with enhanced photosynthesis and lowered pCO2 levels observed in some eutrophic impounded reaches of the Mississippi (Crawford et al., 2016), the Yangtze (Liu et al., 2016), and a Yellow River tributary (Ran et al., 2017)"
You repeat twice the "lower pCO2 here, rephrase, something like "in impoundments phytoplanktonic primary production is favoured (REF…) and pCO2 is lowered"
**<Response> Rephrased (**the lowered $pCO_2$ levels as a consequence of increased primary productivity**).**

L343 what does "taper" mean?
**<Response> Changed (**gradually decrease**).**

L348 "which may be associated with terrestrial DOM components and their microbial transformation products". Is it "may be associated" or does it reveal a degradation of terrestrial DOM to CO2?
**<Response> To specify each indicator's meaning, the sentence has been reformulated (**While HIX and C1/DOC indicate the degree of humification and the proportion of terrestrial DOM components, respectively, C2/DOC represents the proportion of "microbial humic components" in the bulk DOM (Fellman et al., 2010; Parr et al., 2015).**).**

L348 "However, the concurrence in the relatively high levels of pCO2 and DOC moieties of terrestrial origin at some middle reach sites that are less affected by impoundments (e.g., HR 7 and HR8) might have resulted in the observed significant correlations." Confusing sentence. Please rephrase
**<Response> Rephrased (**In contrast, the relatively high levels of $pCO_2$ concurred with strong optical intensities of terrestrial DOM components at some middle reach sites that are less affected by impoundments (e.g., HR 7 and HR8), resulting in the significant correlations between the relatively wide ranges of $pCO_2$ and DOM optical properties.**).**

L358 what is a "low head dam"?
**<Response> More detail has been provided (**low dams constructed for river navigation**).**

L361 "depth-dependent gradient" > vertical gradient?
**<Response> Yes, and changed.**

L363-367 what does d13C-CH4 say about CH4 oxidation?
Conversion of DOC to CH4 is extremely speculative, (in fact here the correlation is indirect, as they both come from urban areas) because methonogenesis is marginal in the water column, and methanogenesis occurs in sediments using sediment OM and not water column DOC.
"based on the coupling between anaerobic organic matter degradation and methanogenesis" what coupling? This is meaningless. In fact, methanogenesis IS anaerobic organic matter degradation.
**<Response> The sentences have been revised to provide a more coherent explanations (**$CH_4$ concentrations in the middle reach exhibited a weak, but significant correlation with DOC concentrations (Fig. 5, Table S4). This correlation may indicate an active methanogenesis in anaerobic reservoir

sediments that is often accompanied by increases in surface water DOC concentrations (Chen et al., 2009; Wang et al., 2017b). It is also possible that some local sources of organic wastes surrounding the reservoirs may have directly discharged wastewater rich in DOC and $CH_4$ (Bergier et al., 2014; Wang et al., 2017b).**).**

L374 "might be explained by the complex interactions between microbial N transformations in the oxygen-rich epilimnion and oxygen-poor hypolimnion" be precise, what processes?, nitrification, denitrification, DRNA, anamox? In addition the fact that N2O does not peak in reservoirs does not mean it is a result of "complex" processes, it might be very simple: no production of N2O…
**<Response> The new sentence reads "**The lack of clear impoundment effects on $N_2O$ concentrations except for one reservoir (HR6; Fig. 2) can be explained by little $N_2O$ production in the other reservoirs or the complex interplay between $N_2O$ production from nitrification and denitrification and $N_2O$ consumption under changing availability of $O_2$**".**

L375 "a suite of related processes" related to what? "in stream metabolism" ok, but heterotrophic or autotrophic, respiration or primary production? Please be more precise.
**<Response> The phrase has been changed to make clear the point of the sentence** (Large increases in GHG concentrations along the lower reach may be a combined result of the net in-stream production and direct inputs from WWTPs.)**.**

L378 and later in the MS: the term "pulse" refer to temporal changes not spatial changes. It is not appropriate for describing local inputs due to urban areas
**<Response> We have replaced the term "pulse" by "pulse-like" or "pulsatile" or "peak concentrations" to indicate some strong localized concentration increases, as often found in the literature.**

L379 "benthic sediment" is a truism
**<Response> Changed to "bottom sediment".**

L384-388 how was this budget calculated? "was estimated to be consumed by phyto…" how? "same reach" do you mean the same river section?
**<Response> More details on the mass balance approach have been provided in the preceding sentences (**Previously we used a mass balance approach based on three cruise underway measurements of $pCO_2$ and DOC and the estimated rates of $CO_2$ outgassing from the same reach of the lower Han River, and additional measurements of $pCO_2$ and DOC at two urban tributaries (TC and JN) to show that the two tributaries JN and TC delivering WWTP effluents accounted for up to 72% of the $CO_2$ concentration measured at a downstream location of the lower reach (Yoon et al., 2017). When the rates of $CO_2$ production, consumption, and outgassing were estimated using the mass balance approach for a section upstream and two sections downstream of the two tributaries lower reach in June 2016 (Yoon et al., 2017),...**. The last sentence has been changed to indicate the mass balance-based estimation (…**the mass balance suggested that the bulk of $CO_2$ delivered by the tributaries might have been consumed by phytoplankton photosynthesis in the downstream section of the lower reach (Yoon et al., 2017).**).**

L390 "By directly measuring d13C in CO2 respired by bacterioplankton across a gradient of streams and lakes in Canada, McCallister and del Giorgio (2008) showed that the production of CO2 through bacterial degradation of terrigenous DOM decreased in sharp contrast to the increasing proportion of algal-derived DOC and CO2 with increasing levels of Chl a" It is hard to find the relevance of this observation on boreal lakes with that here in Korean rivers. If the question here is whether algae release (exudate DOC) that is further mineralized to CO2 by heterotrophic microbes, or does the algae directly respire in the dark, I think you will never answer this question. Out of the scope of the paper.
**<Response> Yes, we agree, but we just wanted to emphasize the relative importance of these two processes by citing this rare finding. We have added a caveat that we need to consider different mechanisms in different climate zones (…** although it would require further research to verify the findings in the boreal freshwaters in temperate and other biomes**).**

L398 "ranges of two gases" please specify which gases
**<Response> Specified.**

L401 "point to" is not correct English
**<Response> Changed to "indicate".**

L405 "In contrast to the lacking or weak correlations indicative of anaerobic CH4 production in the impounded middle reach" I wonder what is a "correlation indicative of anaerobic CH4 production"? please rephrase.
What is a "correlation related to anaerobic metabolism"? please rephrase
**<Response> To avoid any confusion of data interpretation, the sentence has been rephrased to describe the different correlations found in the middle and lower reaches** (In contrast to the nonsignificant correlation between $CH_4$ and DO in the impounded middle reach, $CH_4$ measurements in the lower reach exhibited either a significant negative (DO) or positive correlation (water temperature, C1/DOC, and C2/DOC).)**.**

L408 "eutrophication (indicated by significant positive correlation with PO4)" correlation of what parameter with PO4? Please rephrase
**<Response> Two related sentences have been rephrased:**
In contrast to the nonsignificant correlation between $CH_4$ and DO in the impounded middle reach, $CH_4$ concentrations in the lower reach exhibited a significant negative correlation with (DO) as well as positive correlations with $PO_4^{3-}$, water temperature, C1/DOC, and C2/DOC. As observed in other urbanized river systems (Beaulieu et al., 2015; Smith et al., 2015; Wang et al., 2018), $CH_4$ correlated positively with $PO_4^{3-}$, but negatively with DO, implying that the nutrient enrichment often leading to severe phytoplankton blooms during warm summer months may create favourable conditions for anaerobic methanogenesis in the lower reach that is almost impounded by the two submerged weirs.

In general the ideas contained in this 4.1 section are ok and make sense, but the phrasing in English is hazardous which makes the message confusing
**<Response> In addition to correcting the commented sentences, the entire section has been doulbe checked to enhance clarity.**

L419: 14CDOC in a wide range of global river systems" Please rephrase
**<Response> Changed to "rivers around the world".**

L423: "large fluctuations in d13CDOC along the upper to middle reaches from HR2 to HR11 do not present any consistent longitudinal trend of the stable C isotopic composition" Awkward sentence, please rephrase
L425 you use here the term "autochtonous", do you mean phytoplankton in majority? d13C-DIC of phytoplankton varies a lot depending of the 13C signature of the DIC it has been using.
L426 "which deviated substantially from those of the headwater DOM dominated by allochthonous components". Please be precise, mentioning what isotope and what are the respective values. What's the meaning of "allochtonous component" here?
**<Response> In response to these three comments, the sentence has been split an rephrased (**While $\delta^{13}C_{DOC}$ was highly variable along the upper to middle reaches from HR2 to HR11, the values of $\delta^{13}C_{DOC}$ at the most downstream site HR14 (−20.6‰ and −23.2‰) were distinctively higher than those measured at the forested headwater stream (−28.2‰ on both sampling dates) (Table S6). The DOM optical properties measured at HR14 were also significantly different from the high HIX and low FI values indicating the predominance of soil-derived DOM in the headwater stream (Fig. 3; Table S1). Taken together, the isotopic composition and optical properties of DOM in the lower reach may reflect the downstream addition of DOM components derived from anthropogenic sources such as WWTP effluents ($\delta^{13}C_{DOC}$ around −26‰; Table S6) or plankton biomass (note the wide range of the plankton $\delta^{13}C_{DOC}$.in Fig. 7).)**.**

L428 "In addition, the distinct seasonal differences in isotopic signatures suggest that the age of DOM is generally younger at five mainstem sites across the river basin during the monsoon period (July 2014; modern -590 years B.P.) than in the dry season (May 2015; 180 - 675 years B.P.)" Do the isotopic values

suggest they vary or do they simply show variation? Please rephrase. Provide the values of 14C together with the age.

**<Response> 14C values have been provided in the relevant result section and the rephrased sentence now reads "…**the distinct seasonal differences in $\Delta^{14}C_{DOC}$ across the five mainstem sites illustrate that the age of DOM is…**".**

L433: why "latitudinal" here? Are values plotted versus latitude?
**<Response> Corrected to "longitudinal".**

L434 is WWTP DOC older or younger? Could the downstream trend in 14C originate from selective degradation of young C?
**<Response> The rephrased sentence reads "**The longitudinal increase in DOM age from a modern age to 180 years B.P. at the forested headwater stream to 590–675 years B.P. at the most downstream site may reflect a preferential degradation of young, labile components during riverine DOM transport (Raymond and Bauer, 2001), but also indicates a significant contribution of aged DOM derived from downstream anthropogenic sources; for example, the age of DOM measured at the outlet and WWTP effluents of the urban tributary JN ranged from 765 to 1050 years B.P. (Table S6).**".**

L438 "As suggested by Griffith and Raymond (2011), aged DOM moieties in WWTP effluents (765 - 905 years B.P.; Table S6) may not only leave clear isotopic signatures on DOM in downstream reaches, but also fuel the riverine heterotrophy by providing labile sources for biodegradation" Is it younger or older DOC being degraded? Please reword "may not leave clear"
**<Response> Griffith and Raymond (2011) suggested that even aged DOM from wastewater can provide labile sources. The rephrased sentence reads "…**aged DOM derived from the WWTP effluents (765–905 years B.P.; Table S6) may contain labile materials, which, mixed with other labile components from in-stream sources such as phytoplankton, can fuel the riverine heterotrophy along the lower reach.**".**

L440 "might be related to a complex array of interacting processes such as organic matter degradation, photosynthesis by phytoplankton, and atmospheric gas exchange, which have usually been investigated as determinants of the isotopic composition of riverine DIC consisting of dissolved CO2, bicarbonate, and carbonate " this is too general and confusing. Gas exchange fractionated because 12CO2 degases faster than 13CO2, uptake by phytoplankton fractionate because algae preferentially use 12C, leaving more 13C-DIC, and respiration releases DIC with a signature close to that of the OM source (not fractionation but mixing with 13C depleted DIC). It looks you only refer to Mc Callister and del Giorgio regarding 13C-DIC (or 13C CO2), but their study site is very different from yours
**<Response> As mentioned before, we have rewritten the entire section including the commented sentences and cited more papers on d13C-CO2 (e.g., Campeau et al., 2017) in L 500-515. Briefly summarized, we now compare the observed d13C-CO2 values with d13-DOC values measured at the forest stream and d13C-CO2 values in Swedish forest streams reported by Campeau et al. to emphasize that downstream riverine processes (outgassing and photosynthesis) constrain the downstream enrichment of 13C in CO2 along the lower reach. In the following paragraph, we compare our estimated d13C-DIC values with literature information to explain further processes involved in downstream changes in DIC sources.**

L447 "When the observed values of 13CCO2.. it follows then that other riverine processes than bacterial degradation of plant (predominantly C3 in the studied basin) and algal biomass might be involved in the upward shift of 13CCO2." The interpretation is wrong here: algal biomass must be considered as a fractionating process during primary production rather than an input of DIC from degradation of algae. The production of algal biomass will increase de 13C-CO2 and 13C-DIC and is probably responsible for the observed trend, maybe with water-air isotopic equilibration.
"assuming and enrichment of 10‰o..." this enrichment may vary between what and what values? (I guess between 5 and 15‰o). In fact because you have alkalinity, you should be able to calculate the DIC and the concentration of all DIC species as well as the isotopic composition of each species and the total DIC. See Zhang J., Quay P. D. and Wilbour D. O. (1995) Carbon isotope fractionation during gas–water exchange and dissolution of CO2. Geochim. Cosmochim. Acta 59, 107–114.
**<Response> The estimated d13C-DIC values based on Zhang et al. are now provided in L 522-525.**

L463 what does "flanked" mean here?
**<Response> Changed to "separated".**

L467 Weir will enhance CO2 degassing and increase d13C-DIC
What is a "tributary-specific process"?
DOM from WWTP have old 14C signal because of a small contribution of fossil OM such as fuel and gasoline. However, these compounds represent a very marginal contribution to the total DOC. Thus the 14C signal as a tracer of source is hampered by this contamination. See Marwick et al. 2015
**<Response> As mentioned, we have restricted the use of d14C-DOM to evaluate the wastewater effect on the lower reach DOM.**

L473 why referring to "latitudinal" here? No latitude in figure 8
**<Response> Corrected to "longitudinal".**

L478 "pulsating" or "pulse" refer to temporal dimension. Maybe you mean "hotspots" or "point source" here?
**<Response> As explained before, we have used the term "pulse-like" or "pulsatile" to emphasize very strong, localized peaks, as used in the literature.**

L483 "Down-river concentration decreases and 13C enrichment in CH4 are consistent with the underway measurements conducted using a cavity ring-down spectroscopy(CRDS) along a 15 km reach of the North Creek estuary in Australia, which displayed CH4 concentrations…. respectively (Maher et al., 2013)." Comparing river with an estuary is not necessarily appropriated because in estuaries, mixing with seawater occurs.
**<Response> We cited this paper to compare the opposing patterns of CH4 and its d13C, both of which have rarely been measured simultaneously in rivers. That's why we had to keep the estuarine study. To make the point (CH4 oxidation enriching 13C) clearer, another river study has been added and the related sentences have been rephrased, as follows:**

The contrasting down-river trends of decreasing $CH_4$ concentrations and $^{13}C$ enrichment are consistent with the longitudinal patterns of $CH_4$ concentration and its stable C isotope ratios measured simultaneously in large rivers such as the Amazon River (Sawakuchi et al., 2016) and estuaries (Maher et al., 2013). Sawakuchi et al. (2016) found the increases in $\delta^{13}C_{CH4}$ and the abundance of a genetic marker for methane-oxidizing bacteria (*pmoA*) in waters with lower $CH_4$ concentrations across the mainstem and tributaries of the Amazon. They used stable isotopic mass balances of $CH_4$ in the water column and estimated that 17–100% of $CH_4$ produced in the riverbed sediment may be oxidized during transport through water column to the atmosphere. During cruise expeditions employing a cavity ring-down spectroscope (CRDS) along a 15 km reach of the North Creek estuary in Australia, Maher et al. (2013) observed increasing $\delta^{13}C$ values from −61.07 to −48.62‰ in contrast to large decreases in $CH_4$ concentrations from 74 to 2 nmol in the downstream direction. $CH_4$ oxidation was suggested as the primary driver of downstream increases in $\delta^{13}C$ in the studied estuary with relatively low levels of anthropogenic pollution (Maher et al., 2013). The down-river patterns of $CH_4$ concentration and isotopic composition observed in this study also suggest that $CH_4$ oxidation in the well-mixed, shallow water, in combination with physical evasion to the atmosphere, may efficiently remove $CH_4$ derived from multiple sources including the urban tributaries enriched in $CH_4$ and riverbed sediments affected by the eutrophic water and frequent phytoplankton blooms.

L489. You did not measure d15N-N2O so no need for this last sentence
**<Response> The sentence has been removed.**

Section 4.2 needs profound revision making clearer the question of 13C CO2 versus 13C DIC and the story of old/young DOM
**<Response> The entire section has been revised thoroughly. Regarding 13DIC, please refer to our earlier responses to the major comment on the same issue.**

L493: "5.implications" of what? strange title for a last section

Mentioning the river continuum concept here is too late. In addition, one may say that your study does not show a "limited validity" of the concept, to the contrary, it suggest that anthropogenic actions might alter it, but the concept continues being valid if we consider these impact as artificial.
**<Response> A more specific title (**Implications for integrative concepts and future research**) is used for this final discussion section emphasizing the necessity of novel concepts reflecting anthropogenic perturbations. The river continuum concept is described in Introduction. Please note that we have revised the text so that we emphasize, not "limited validity", but "anthropogenic alterations".**

498 "Borges and Abril 2011" is for estuaries, not adapted here
**<Response> Removed.**

L499 "eutrophic reaches receiving wastewater" do you simply mean river sections receiving WW?
**<Response> Rephrased (**eutrophic waters polluted by wastewater**)**

L500 awkward sentence, please rephrase
**<Response> As explained earlier, we have removed the sentence and instead reformulated the following sentences to focus more on the implications of our findings.**

L506 "Reach-specific significance levels observed for the correlations between GHGs and DOC or its optical properties (Fig. 5; Table S4) imply that…" awkward sentence, please rephrase
the relative contributions of autochthonous production and external supplies of GHGs derived from WWTP effluents
L510, "pulse" refers to time, not to space
**<Response> Now "peak" used.**

L511 replace biodegradation by respiration
**<Response> Replaced**

L514 "gas-specific sets of significant correlations between gas concentrations and related water quality measurements shown for the lower reach" such sentence says basically nothing. What parameters, what gas?
**<Response> Rephrased (**The different significance levels established between the three GHGs and measured nutrients implied some different roles that those nutrients may play in the production of each gas in the eutrophic lower reach**).**

L517 algal productivity will NOT favour CO2 production, to the contrary, it favours consumption
**<Response> The sentence has been rephrased to describe the net effect of excess algal growth and subsequent development of anaerobic conditions (…**enhanced phytoplankton growth and anaerobic metabolic activity in the eutrophic reach often plagued by phytoplankton blooms may result in the net positive effect on the production of both $CO_2$ and $CH_4$ despite the immediate negative effect of algal uptake on the surface water level of dissolved $CO_2$**).**

L522 "Urban tributary effects on metabolic processes in the eutrophic lower reach were also reflected in the lower-rech values of d13C in CO2 and CH4 resembling those measured in the tributaries". This sentence is almost impossible to understand. What metabolic processes are you referring to?
**<Response> This secondary information has been removed not to cause any unnecessary confusion.**

L525 what "metabolic processes"?
Strange the term "hotspot" only appears here
**<Response> New words have been selected to compare two different sources (**the relative contributions of autochthonous production and external supplies of GHGs derived from WWTP effluents**).**

Figures: why are time courses of monthly monitoring shown only in Fig6 and not before in the MS?
**<Response> Monthly data are presented in the beginning section (Tables 1 and S1 and Fig. S2). Because there is no clear temporal pattern (Fig. S2) and it is simply too complex to show the three gas measurements at six sites in a single figure, we opted for the tables and the summary figure.**

Fig2 I could not see the "Dashed horizontal lines". What is the yellow area?
**<Response> An updated version has been included. Dashed lines indicating atmospheric equilibrium have been added and the yellow shade has been described in the figure caption.**